# Tracking the $^{10}$Be-$^{26}$Al source-area signal in sediment-routing systems of arid central Australia

Martin Struck[1], John D. Jansen[2], Toshiyuki Fujioka[3], Alexandru T. Codilean[1], David Fink[3], Réka-Hajnalka Fülöp[1,3], Klaus M. Wilcken[3], David M. Price[1], Steven Kotevski[3], L. Keith Fifield[4], and John Chappell[4]

[1]School of Earth and Environmental Sciences, University of Wollongong, Wollongong 2522, Australia
[2]Department of Geoscience, Aarhus University, 8000 Aarhus C, Denmark
[3]Australian Nuclear Science and Technology Organisation, Lucas Heights 2234, Australia
[4]Research School of Earth Sciences, Australian National University, Canberra 0200, Australia

**Correspondence:** Martin Struck (ms646@uowmail.edu.au)

**Abstract.** Sediment-routing systems continuously transfer information and mass from eroding source areas to depositional sinks. Understanding how these systems alter environmental signals is critical when it comes to inferring source-area properties from the sedimentary record. We measure cosmogenic $^{10}$Be and $^{26}$Al along three large sediment-routing systems ($\sim$100,000 km$^2$) in central Australia with the aim of tracking downstream variations in $^{10}$Be-$^{26}$Al inventories and to identify the factors responsible for these variations. By comparing 56 new cosmogenic $^{10}$Be and $^{26}$Al measurements in stream sediments with matching data (n = 55) from source areas, we show that $^{10}$Be-$^{26}$Al inventories in hillslope bedrock and soils set the benchmark for relative downstream modifications. Lithology is the primary determinant of erosion-rate variations in source areas and despite sediment mixing over hundreds of kilometres downstream a distinct lithological signal is retained. Postorogenic ranges yield catchment erosion rates of $\sim$6–11 m m.y.$^{-1}$ and silcrete-dominant areas erode as slow as $\sim$0.2 m m.y.$^{-1}$. $^{10}$Be-$^{26}$Al inventories in stream-sediments indicate that cumulative-burial terms increase downstream to mostly $\sim$400–800 k.y. and up to $\sim$1.1 m.y. The magnitude of the burial signal correlates with increasing sediment cover downstream and reflects assimilation from storages with long exposure histories, such as alluvial fans, desert pavements, alluvial plains, and aeolian dunes. We propose that the tendency for large alluvial rivers to mask their $^{10}$Be-$^{26}$Al source-area signal differs according to geomorphic setting. Signal preservation is favoured by i) high sediment supply rates, ii) high mean runoff, and iii) a thick sedimentary basin pile. Conversely, signal masking prevails in landscapes of i) low sediment supply and ii) juxtaposition of sediment storages with notably different exposure histories.

## 1 Introduction

Landscapes are continuously redistributing mass in response to tectonic and climatic forcing. A suite of surface processes achieves this redistribution at rates fast and slow, modifying landscapes while routing particles from erosional source areas to depositional sinks (Allen, 2008). Rapid, short-term transport ($<10^1$ yr) allows for direct monitoring whereas indirect methods such as geochemical-isotopic tracing or mathematical modelling become necessary beyond historical timescales ($>10^2$ yr)

(Allen, 2008; Romans et al., 2016). Longer timescales are also relevant to the making of the geological record, which forms the basis of how we understand the narrative of Earth's history (Allen, 2008). The typical approach involves a classic inverse problem whereby attributes of the source area are inferred retrodictively from the geological record. What is inevitably missed, however, is the range of surface processes and dynamics that particles undergo between source and sink. Considering that particles in transit carry an environmental signal of their source area (Romans et al., 2016), this signal is liable to become obscured en route by the intrusion of 'noise', which we take to mean 'any modification of the primary signal of interest' (Romans et al., 2016, p. 7). Indeed, the ratio of signal to noise is the chief limiting factor for accurately inferring source-area information—in addition to the rudimentary understanding of how environmental signals are propagated through sediment-routing systems over $>10^5$ yr timescales (Romans et al., 2016).

Modern sediment-routing systems provide the opportunity to track changes in the source-area signal with distance downstream. Arid lowland regions, our focus here, offer insights to the propagation of source-area signals in landscapes of low geomorphic activity. Shield and platform terrain under aridity sustains some of the slowest known erosion rates (Portenga and Bierman, 2011; Struck et al., 2018). These low-relief landscapes are characterised by slow sediment production coupled with slow and intermittent sediment supply to surrounding basins. The typically slow rate of crustal deformation means limited accommodation space, resulting in thin and discontinuous sedimentary records (Armitage et al., 2011). Aridity imposes a strongly episodic character to the sediment-routing system. Infrequent rainfall and stream discharge leads to lengthy and irregular intervals of sediment storage in vast low-gradient river systems. It has been suggested that long hiatuses in sediment transfer may increase the potential for diminishing the signal to noise ratio, but this notion is yet to be tested comprehensively.

Terrestrial cosmogenic nuclides are produced by secondary cosmic rays interacting with minerals in the upper few metres of Earth's surface (Gosse and Phillips, 2001), hence they are powerful tools for tracking particle trajectories in the sediment-routing system (Nichols et al., 2002; Matmon et al., 2003; Heimsath et al., 2005; Jungers et al., 2009; Anderson, 2015). Radionuclides, such as $^{10}$Be and $^{26}$Al, are used widely to quantify the erosional dynamics of landscapes on $10^3$–$10^6$ yr timescales (Lal, 1991; McKean et al., 1993; Brown et al., 1995; Granger et al., 1996). Yet, the source-area signal of interest is most often limited to identifying differential erosion rates across a range of spatial scales. For instance, $^{10}$Be abundances in bedrock indicate a point-specific weathering rate and in fluvial sediment $^{10}$Be is used to derive a spatially-averaged catchment erosion rate (Granger et al., 1996). Both approaches entail assumptions that frame how the source-area signal is viewed. Bedrock erosion rate calculations assume steady long-term exhumation (Lal, 1991), and catchment-averaging assumes that the fluvial sediment sample is a representative amalgam of particles generated across the entire catchment (Brown et al., 1995; Bierman and Steig, 1996; Granger et al., 1996). Heterogeneity in the sample may arise due to particles sourced disproportionately from i) faster eroding areas, such as landslides, or ii) landforms that contain notably longer exposure histories, such as ancient alluvium and aeolian dune fields—either case introduces noise that can bias erosion rate calculations (Granger et al., 1996; Norton et al., 2010). A further key assumption is that samples (including bedrock) have not experienced long-term burial. However, in this case, the noise introduced by burial produces some interesting and exploitable effects. By measuring a nuclide pair with differing radioactive decay rates (e.g. $^{10}$Be-$^{26}$Al) the cumulative burial history can be explicitly tracked by the gradual deviation in the initial production ratio of the two nuclides (Granger and Muzikar, 2001).

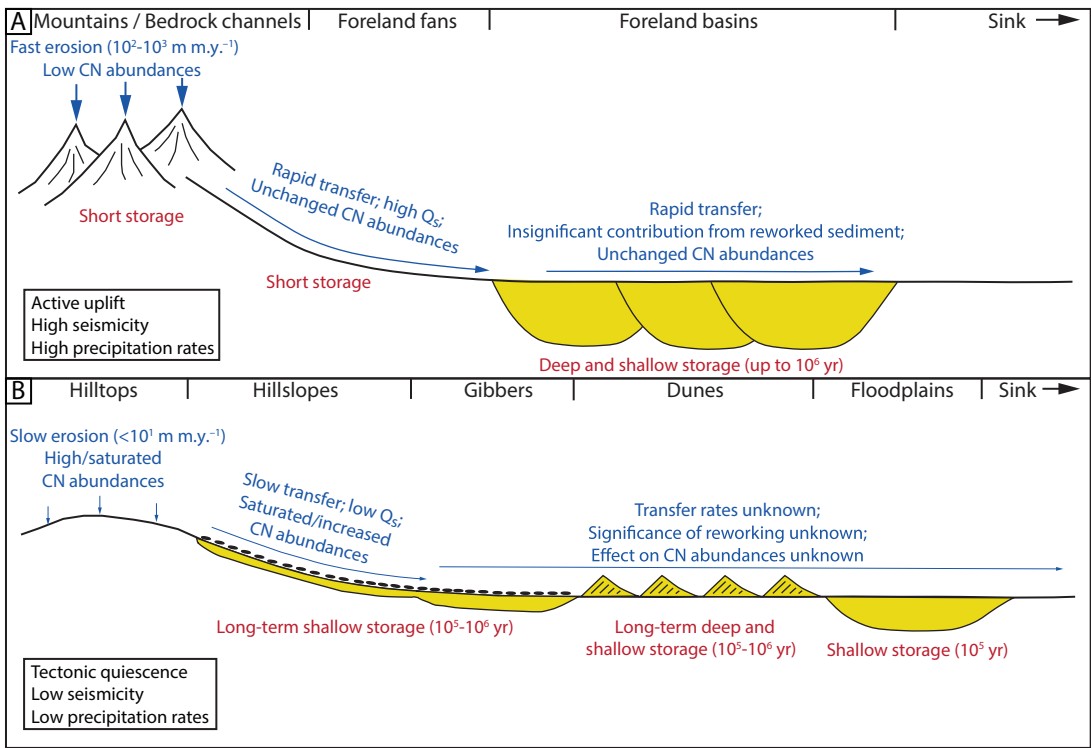

**Figure 1.** Two schematic limit cases of sediment-routing systems (modified after Romans et al., 2016) showing down-system trends from (A) high-relief, tectonically-active mountains with humid climate, and (B) low-relief, postorogenic setting with arid climate. Blue script denotes relative rates of erosion and material transfer and their effects on the cosmogenic nuclide inventory ($Q_s$ is sediment flux). Red script denotes relative burial depths (shallow <10 m, deep >10 m) and storage durations. Yellow shading indicates significant sediment storage.

Several studies apply this approach to understand how [10]Be-[26]Al source-area signals are modified during transit through the sediment-routing system and suggest two broad limit cases: i) [10]Be-[26]Al source-area signals remain largely unmodified from source to sink (Clapp et al., 2000, 2001, 2002; Wittmann et al., 2011; Hippe et al., 2012; Wittmann et al., 2016), or ii) [10]Be-[26]Al source-area signals become significantly obscured with distance downstream (Bierman et al., 2005; Kober et al., 2009; Hidy et al., 2014). Much remains to be understood about the governing controls on the alteration or otherwise of the source-area signal. The heavy emphasis to date has been with studies of sediment-routing systems conveying a source-area signal specific to rapidly eroding mountain belts (Fig. 1A). It seems likely that the transmission of source-area signals will differ across the much larger proportion of Earth's terrain that is low-relief, tectonically-passive, and subject to much lower rates of geomorphic activity (Fig. 1B).

Here we focus upon the shield and platform landscapes that characterise much of the arid interior of Australia, as well as large portions of other Gondwana segments such as Africa, India, and South America. We measure abundances of cosmogenic [10]Be and [26]Al in fluvial sediment within rivers draining source areas for which we have established the [10]Be-[26]Al source-area signal from bedrock and hillslope systems (Struck et al., 2018), and we supplement those with four thermoluminescence

ages on floodplain sediments. Tracking the source-area signal through three large sediment-routing systems via a nested set of samples, we investigate: 1) downstream variations in source-area [10]Be-[26]Al inventories; 2) the factors that modify the [10]Be-[26]Al source-area signal; and 3) how changes in [10]Be-[26]Al inventories along the course of these streams affect erosion rate calculations. We conclude by reflecting upon the implications of our findings for a source to sink understanding of the tempo

of change in arid, shield and platform landscapes.

## 2 Sediment-routing and timescales of landscape evolution in central Australia

Western tributaries of the Eyre Basin: the Finke, Macumba, and Neales rivers drain >100,000 km$^2$ of the arid continental interior (Fig. 2). Low postorogenic ranges of early Palaeozoic and Proterozoic rocks (Fig. 3A) and Cenozoic silcrete-duricrust tablelands (Fig. 3B) serve as the major sources of sediment and runoff for the sediment-routing systems. These traverse hun-

dreds of km of low-relief stony soil mantles (Fig. 3C), alluvial plains, and aeolian dune fields before reaching the depositional sink, Lake Eyre (Fig. 1B). The western Eyre Basin experiences mean temperatures of ∼20°C and mean rainfall of ∼280–130 mm yr$^{-1}$ with extreme interannual variation. Vegetation is sparse: chenopod shrublands and tussock grasslands predominate in the south and mixed open woodland and spinifex in the north, reflecting the northward transition from winter to summer rainfall dominance (Australian Bureau of Meteorology: http://www.bom.gov.au/climate/). Significant flow in the western trib-

utaries is generated mainly by summer rainfall today (Kotwicki, 1986; Costelloe, 2011). Finke River flows have not reached Lake Eyre in historical times (McMahon et al., 2008), but large floods along the Neales have done so repeatedly in more recent years (Kotwicki, 1986; Kotwicki and Isdale, 1991). Periodic high-magnitude flooding in Eyre Basin rivers triggered phases of deposition and incision recorded in fluvial and lacustrine sediments over >300 k.y. (Nanson et al., 1992; Croke et al., 1999; Nanson et al., 2008; Cohen et al., 2012, 2015).

[10]Be-derived erosion rates in the Eyre Basin are among the slowest known (Portenga and Bierman, 2011). Rates are <5–10 m m.y.$^{-1}$ for bedrock outcrops (Fujioka, 2007; Heimsath et al., 2010; Struck et al., 2018) and 5–20 m m.y.$^{-1}$ at catchment-scale (Bierman et al., 1998; Heimsath et al., 2010). The slow evolution of the central Australian landscape is a function of low relief due to restricted tectonic uplift (Sandiford, 2002; Sandiford et al., 2009; Jansen et al., 2013) combined with intensified aridity since the Miocene (Bowler, 1976; McGowran et al., 2004; Martin, 2006; Fujioka and Chappell, 2010). Ongoing intra-

plate tectonic deformation is driven by far-field compressive stresses (Sandiford et al., 2004; Hillis et al., 2008; Waclawik et al., 2008; Sandiford and Quigley, 2009) together with dynamic processes beneath the lithosphere, which have caused long-wavelength deformation on the order of hundreds of metres in vertical amplitude (Sandiford et al., 2009). Clear evidence of rapid Neogene to modern uplift occurs on the southern fringe of the Eyre Basin in the Flinders Ranges and at Billa Kalina (Callen and Benbow, 1995; Sandiford et al., 2009; Quigley et al., 2010).

In a comprehensive assessment of [10]Be-[26]Al abundances in bedrock and soil-mantled source areas in the Eyre Basin, Struck et al. (2018) quantify soil residence times of ∼0.2–2 m.y. and possibly longer at the top of the sediment-routing system. Long residence times and slow hillslope evolution arise from the lack of fluvial incision associated with widespread base-level stability and the long-lasting development of stony soil mantles, also known as desert pavement (Mabbutt, 1977; Wells et al.,

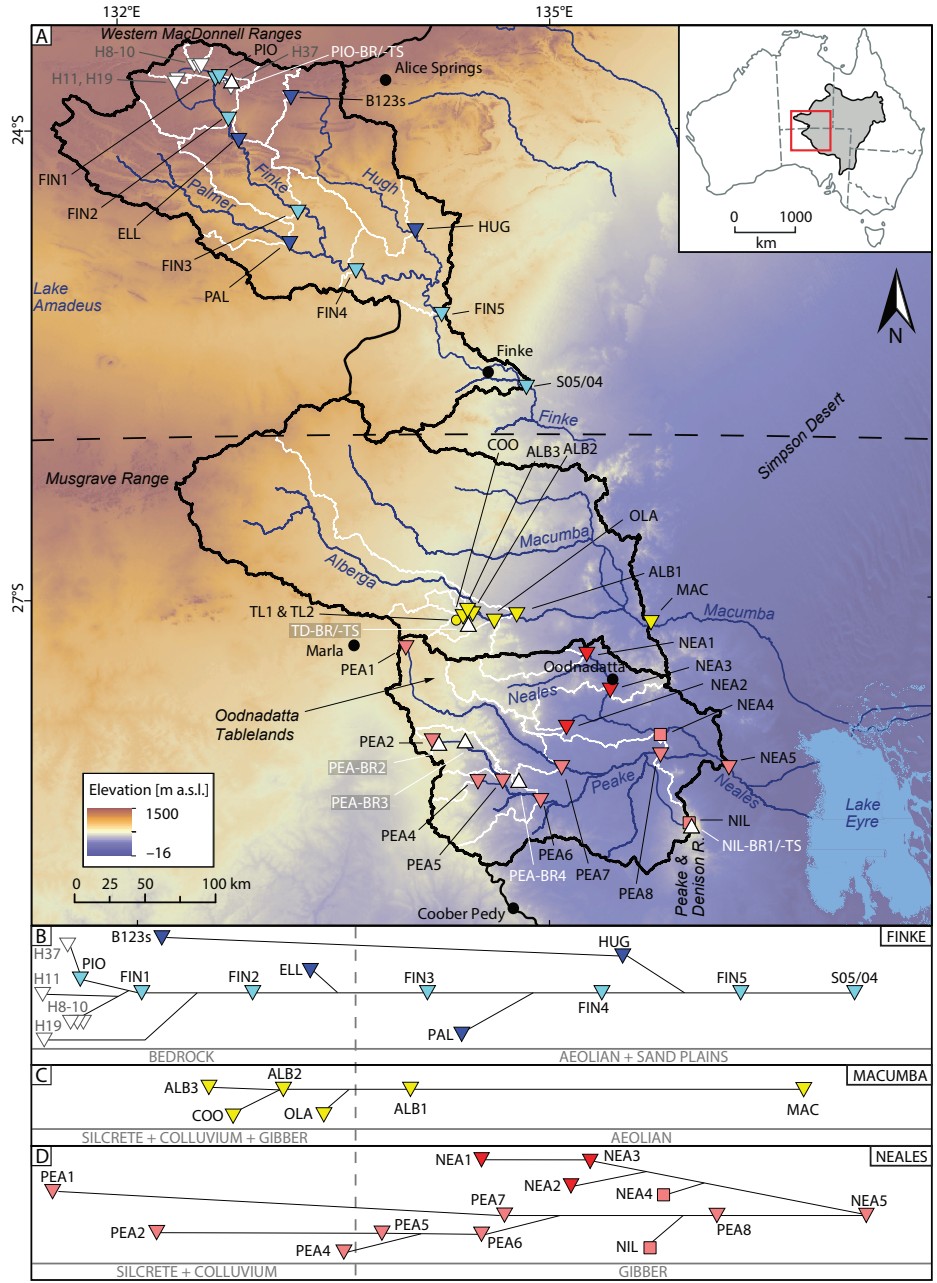

**Figure 2.** A) Three study catchments in the western Eyre Basin, showing stream sediment samples (downward-pointing triangles and squares), bedrock and hillslope samples (upward-pointing white triangles), and thermoluminescence samples (yellow circle). Finke: trunk stream (light blue) and tributaries (dark blue – this study, white – Heimsath et al., 2010), Macumba (yellow), Neales: Neales subcatchment (dark red triangles), Peake subcatchment (light red triangles), streams draining the Peake and Denison Range (light red squares). Eyre Basin (inset: 1.1 million km$^2$) boundaries and outer catchment boundaries (bold black), subcatchment boundaries (white); rivers (blue), towns (black dots), state border (dashed black line). B, C, D) Schematic sediment-routing networks of the Finke, Macumba, and Neales, subdivided according to overall terrain type.

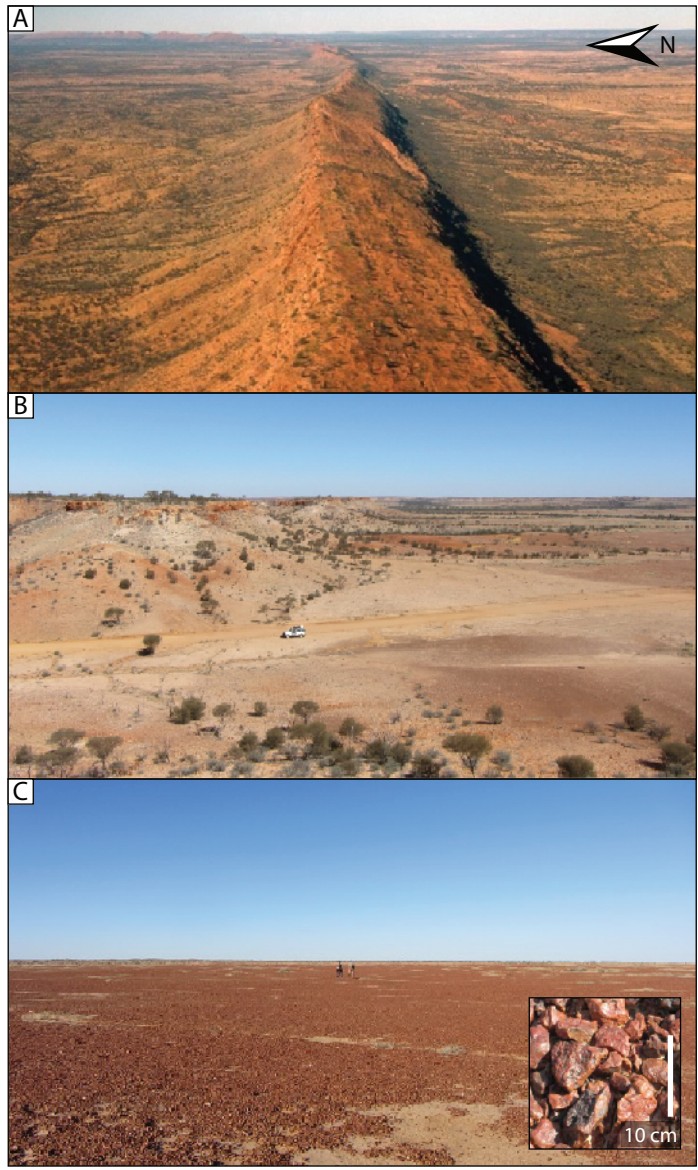

**Figure 3.** A) Typical strike ridges of steeply-inclined strata of the Western MacDonnell Ranges separated by sediment-mantled terrain, Finke River headwaters (Photo: Geoscience Australia). B) Flat-topped, silcrete-capped mesas of the Oodnadatta Tablelands, western headwaters of the Neales River (note 4WD vehicle for scale). C) Gibber-covered palaeo-alluvial plains in the lower Neales catchment, with distant mesas on the skyline (note persons for scale). Inset shows desert-varnished surface silcrete pebbles.

1995; Fujioka et al., 2005; Matmon et al., 2009). Hillslope dynamics reflect 'top-down' evolution (Montgomery, 2003) with slow rates of authigenic soil production and downslope transport resulting in low connectivity with stream channels (Egholm et al., 2013). Inputs of aeolian dust to soils since at least 0.2 m.y. and up to 1 m.y. or more lie stabilised beneath stony soil mantles developed over the past ∼650 k.y. Nuclide abundances in these source-area materials are naturally very high (Fujioka et al., 2005; Fisher et al., 2014; Struck et al., 2018), but low $^{26}$Al/$^{10}$Be ratios also suggest a complex history of either cyclic exposure-burial and/or non-steady exhumation on these hillslopes over timescales of $10^5$ to $10^6$ yr (Struck et al., 2018).

We set out to test three potential sediment transfer scenarios: 1) $^{10}$Be-$^{26}$Al inventories remain unmodified downstream due to fast ($\ll 10^5$ yr) sediment transfer and negligible external input; 2) nuclide abundances increase downstream while $^{26}$Al/$^{10}$Be ratios remain constant, which indicates long-term ($\gg 10^5$ yr) near-surface particle trajectories, or input from nuclide-rich, burial-free sediment sources; 3) nuclide abundances decrease downstream, suggesting significant radioactive decay during slow sediment transfer with lengthy burial intervals (Granger et al., 1996; Granger and Muzikar, 2001; Schaller et al., 2004), or input from nuclide-poor, long-buried sources.

## 3   Methods

We used 1 arc-second digital elevation data from the Shuttle Radar Topographic Mission (SRTM) to analyse elevation, slope, and mean relief of area upstream of each sediment sample measured for $^{10}$Be-$^{26}$Al (Table 1). Mean catchment relief was calculated via smoothing with a circular kernel of 2.5 km radius. Precipitation data derive from gridded (5 km) mean annual precipitation 1911–2000 (Australian Bureau of Meteorology: http://www.bom.gov.au/climate/). Analysis of surface geology is based on a digital 1:1 million surface geology map of Australia (Raymond et al., 2012) and 1:250,000 map sheets for additional details. Bedrock and depositional landforms were sorted into seven different classes: exposed bedrock (no silcrete), exposed silcrete, colluvium cover, gibber cover (desert pavement), aeolian cover, sand plains, and alluvium. Of this group, the first three classes were assigned to the bedrock-hillslope domain and the latter four were assigned to the sediment cover domain.

### 3.1   Cosmogenic nuclide analyses

We collected 29 samples of sandy bed material throughout the Finke (n = 11), Macumba (n = 6), and Neales (n = 13) drainage networks (Fig. 2; Table 2)—in addition to 55 $^{10}$Be and $^{26}$Al measurements from bedrock summits and soil mantles in the low-order subcatchments (Struck et al., 2018). Quartz isolation and Be and Al extraction were conducted on the 250–500 μm size fraction of sediment and crushed bedrock samples at the University of Wollongong and at the Australian Nuclear Science and Technology Organisation using standard methods of HF/HNO$_3$ (Kohl and Nishiizumi, 1992), hot phosphoric acid (Mifsud et al., 2013), and ion chromatography (Child et al., 2000). Be and Al isotope ratios were measured on the ANTARES and SIRIUS Accelerator Mass Spectrometers (AMS) (Fink and Smith, 2007; Wilcken et al., 2017) and normalised to standards KN-5-2 or KN-5-3 (Be) (Nishiizumi et al., 2007), and KN-4-2 (Al) (Nishiizumi, 2004) (Table 2). Uncertainties for the final $^{10}$Be and $^{26}$Al abundances (Table 2) include AMS measurement uncertainties, 2% (Be) and 3% (Al) standard reproducibility, 1% uncertainty in the Be spike concentration, and 4% uncertainty in the inductively coupled plasma optical emission spec-

troscopy (ICP-OES) Al measurements, in quadrature. Erosion rates and apparent burial ages are calculated with CosmoCalc 3.0 (Vermeesch, 2007), using time-independent scaling (Stone, 2000) and production mechanisms based on Granger and Muzikar (2001) to give a sea-level high-latitude (SLHL) spallation production rate for $^{10}$Be of 4.18 atoms g$^{-1}$ yr$^{-1}$ (Vermeesch, 2007). We assume a $^{10}$Be half-life of 1.387 $\pm$ 0.012 m.y. (Chmeleff et al., 2010; Korschinek et al., 2010), $^{26}$Al half-life of 0.705 $\pm$

5   0.024 m.y. (Norris et al., 1983) and $^{26}$Al/$^{10}$Be surface production ratio of 6.75 (Balco et al., 2008). Six samples (UHugh199, -299, -399, -499, Be122p, and Be123s; Table 2) were measured for $^{10}$Be at the Australian National University (ANU) Heavy Ion Accelerator Facility (Fifield et al., 2010; see Table 2 for details).

## 3.2 Thermoluminescence dating

With the aim of gauging the burial age of floodplain sediments flanking some of our study channels, we collected four samples

10  for thermoluminescence (TL) dating in the upper reaches of the Macumba catchment (Fig. 2A): one from a borrow pit at 125 cm depth (TL2-125); the other three (TL1-40, -100, -160) in a depth profile (40, 100, 160 cm depth) from a similar pit close by (Table A1). All samples were analysed at the University of Wollongong following Shepherd and Price (1990).

Table 1: Catchment characteristics.

| Sample ID | Distance to outlet[a,b] [km] | Catchment size[a] [km²] | Distance from divide[a,c] [km] | Elevation[a] Mean[d] [m] | Elevation Min [m] | Elevation Max [m] | Slope[a] Median [°] | Slope $Q_1$-$Q_3$ [°] | Slope Kurtosis[e] [-] | Total relief[a] [m] | Average local relief[a,f] [m] | Rainfall[d,g] [mm yr⁻¹] | Exposed bedrock without silcrete[h] [%] | Exposed silcrete[h] [%] | Colluvium cover[h] [%] | Gibber cover[h] [%] | Dunes/ aeolian cover[h] [%] | Sand/ plains cover[h] [%] | Alluvium cover[h] [%] |
|---|---|---|---|---|---|---|---|---|---|---|---|---|---|---|---|---|---|---|---|
| *FINKE catchment* | | | | | | | | | | | | | | | | | | | |
| H11[i] | 607.2 | 0.9 | 1.5 | 812 ± 6 | 798 | 827 | 4 | 3-6 | 0.0 | 29 | 108 | 281 ± 0 | 100 | 0.0 | 0.0 | 0.0 | 0.0 | 0.0 | 0.0 |
| H19[i] | 592.9 | 10.0 | 4.7 | 791 ± 24 | 747 | 873 | 5 | 3-7 | 2.1 | 126 | 141 | 279 ± 0 | 92.3 | 0.0 | 0.0 | 0.0 | 0.0 | 0.0 | 7.7 |
| H10[i] | 581.5 | 211.3 | 29.0 | 820 ± 61 | 674 | 1351 | 5 | 2-13 | 6.3 | 677 | 224 | 277 ± 3 | 91.6 | 0.0 | 7.8 | 0.0 | 0.0 | 0.0 | 0.6 |
| H8[i] | 590.1 | 0.6 | 1.8 | 1061 ± 164 | 783 | 1350 | 48 | 35-61 | 0.5 | 567 | 683 | 272 ± 1 | 100 | 0.0 | 0.0 | 0.0 | 0.0 | 0.0 | 0.0 |
| H9[i] | 579.3 | 0.5 | 1.9 | 972 ± 161 | 703 | 1255 | 41 | 28-53 | -0.5 | 552 | 681 | 272 ± 1 | 100 | 0.0 | 0.0 | 0.0 | 0.0 | 0.0 | 0.0 |
| H37[i] | 572.1 | 0.2 | 0.8 | 771 ± 50 | 701 | 849 | 22 | 13-32 | -0.2 | 148 | 186 | 258 ± 0 | 100 | 0.0 | 0.0 | 0.0 | 0.0 | 0.0 | 0.0 |
| PIO | 562.1 | 98.1 | 26.5 | 777 ± 82 | 649 | 1128 | 4 | 2-10 | 4.5 | 479 | 260 | 262 ± 3 | 98.2 | 0.0 | 0.0 | 0.0 | 0.0 | 0.0 | 1.8 |
| FIN1 | 556.5 | 1545.6 | 69.0 | 802 ± 93 | 625 | 1370 | 3 | 2-7 | 10.4 | 745 | 240 | 272 ± 9 | 87.7 | 3.1 | 6.5 | 0.0 | 0.0 | 0.0 | 2.7 |
| FIN2 | 519.0 | 4016.2 | 117.1 | 761 ± 96 | 568 | 1370 | 2 | 1-5 | 14.2 | 802 | 163 | 270 ± 10 | 63.8 | 1.3 | 2.4 | 0.0 | 0.0 | 22.9 | 9.6 |
| B123s | 490.5 | 6.7 | 4.7 | 845 ± 33 | 763 | 937 | 9 | 1-14 | -0.1 | 174 | 193 | 250 ± 1 | 100 | 0.0 | 0.0 | 0.0 | 0.0 | 0.0 | 0.0 |
| ELL | 481.7 | 1613.6 | 108.9 | 710 ± 111 | 526 | 1285 | 2 | 1-5 | 13.4 | 759 | 166 | 251 ± 6 | 69.3 | 0.0 | 0.9 | 0.0 | 0.0 | 19.8 | 9.8 |
| FIN3 | 381.1 | 8649.9 | 251.9 | 691 ± 132 | 420 | 1370 | 2 | 1-6 | 11.1 | 950 | 158 | 257 ± 16 | 62.8 | 1.0 | 2.7 | 0.0 | 2.7 | 19.9 | 10.9 |
| PAL | 352.7 | 7252.2 | 274.9 | 638 ± 119 | 414 | 992 | 2 | 1-6 | 11.6 | 578 | 132 | 253 ± 19 | 59.6 | 0.2 | 2.9 | 0.0 | 6.1 | 13.9 | 17.3 |
| FIN4 | 273.6 | 20625.9 | 359.4 | 617 ± 148 | 356 | 1370 | 2 | 1-5 | 14.7 | 1014 | 124 | 245 ± 25 | 52.9 | 0.6 | 2.1 | 0.0 | 9.2 | 21.4 | 13.8 |
| HUG | 249.2 | 6857.7 | 258.3 | 573 ± 122 | 365 | 1224 | 1 | 1-2 | 24.7 | 859 | 101 | 239 ± 12 | 39.8 | 0.9 | 1.6 | 0.0 | 2.5 | 47.8 | 7.4 |
| FIN5 | 131.8 | 31706.6 | 501.2 | 576 ± 154 | 287 | 1370 | 1 | 1-3 | 18.8 | 1083 | 110 | 238 ± 25 | 45.7 | 0.8 | 1.7 | 0.0 | 14.2 | 26.2 | 11.4 |
| S05/04 | 0.0 | 38368.7 | 533.0 | 539 ± 164 | 221 | 1370 | 1 | 1-2 | 22.7 | 1149 | 102 | 228 ± 26 | 41.3 | 0.7 | 3.3 | 0.1 | 17.7 | 25.6 | 11.3 |
| *MACUMBA catchment* | | | | | | | | | | | | | | | | | | | |
| COO | 198.5 | 238.5 | 26.8 | 270 ± 30 | 219 | 340 | 1 | 1-2 | 4.8 | 121 | 63 | 171 ± 3 | 0.4 | 35.1 | 25.8 | 32.1 | 0.0 | 0.0 | 6.6 |
| ALB3 | 196.6 | 243.6 | 42.0 | 268 ± 29 | 216 | 382 | 1 | 1-2 | 12.9 | 166 | 62 | 175 ± 4 | 0.0 | 12.7 | 39.5 | 42.8 | 0.0 | 0.0 | 5.0 |
| ALB2 | 190.1 | 1350.5 | 82.7 | 289 ± 42 | 211 | 408 | 1 | 1-2 | 7.5 | 197 | 61 | 176 ± 6 | 0.6 | 30.8 | 23.7 | 37.3 | 0.4 | 2.0 | 5.2 |
| OLA | 162.6 | 792.6 | 81.8 | 268 ± 37 | 193 | 369 | 1 | 0-2 | 9.3 | 176 | 54 | 170 ± 5 | 0.0 | 21.9 | 13.1 | 57.8 | 0.0 | 0.0 | 7.2 |
| ALB1 | 141.5 | 14089.1 | 398.6 | 418 ± 113 | 184 | 811 | 1 | 0-2 | 48.3 | 627 | 39 | 197 ± 17 | 14.7 | 7.5 | 36.2 | 10.2 | 8.9 | 16.3 | 6.2 |
| MAC | 0.0 | 39024.0 | 527.5 | 322 ± 131 | 94 | 811 | 1 | 0-2 | 43.2 | 717 | 33 | 180 ± 22 | 9.4 | 15.9 | 22.3 | 10.6 | 14.8 | 18.6 | 8.4 |
| *NEALES catchment* | | | | | | | | | | | | | | | | | | | |
| PEA1 | 430.3 | 8.3 | 5.9 | 355 ± 9 | 333 | 368 | 1 | 1-2 | -0.3 | 35 | 37 | 179 ± 0 | 0.0 | 75.5 | 24.5 | 0.0 | 0.0 | 0.0 | 0.0 |
| PEA2 | 367.4 | 173.4 | 24.7 | 281 ± 6 | 263 | 303 | 0 | 0-1 | 2.2 | 40 | 15 | 148 ± 4 | 0.0 | 3.2 | 91.5 | 5.3 | 0.0 | 0.0 | 0.0 |
| PEA4 | 299.6 | 460.9 | 52.1 | 259 ± 24 | 184 | 313 | 1 | 0-2 | 3.7 | 129 | 32 | 137 ± 5 | 19.8 | 37.1 | 24.6 | 0.6 | 0.4 | 14.1 | 3.8 |
| PEA5 | 279.3 | 1412.6 | 106.2 | 248 ± 36 | 155 | 316 | 1 | 1-2 | 20.9 | 161 | 40 | 139 ± 7 | 14.8 | 14.5 | 20.1 | 37.1 | 0.0 | 5.7 | 7.8 |
| NEA1 | 241.2 | 963.2 | 107.3 | 207 ± 37 | 135 | 314 | 1 | 1-2 | 6.8 | 179 | 30 | 158 ± 3 | 12.1 | 11.2 | 12.8 | 50.2 | 0.0 | 7.3 | 6.4 |
| PEA6 | 231.0 | 4181.7 | 149.1 | 226 ± 46 | 118 | 316 | 1 | 1-2 | 29.5 | 198 | 37 | 134 ± 8 | 18.1 | 18.2 | 13.4 | 26.4 | 0.0 | 14.1 | 9.8 |
| NEA2 | 200.6 | 173.8 | 45.3 | 187 ± 35 | 124 | 261 | 1 | 1-3 | 10.8 | 137 | 44 | 145 ± 4 | 17.8 | 11.2 | 12.1 | 44.6 | 0.0 | 0.0 | 14.3 |

*Continued on next page.*

*Table 1 continued from previous page.*

| Sample ID | Distance to outlet [a,b] [km] | Catchment size [a] [km²] | Distance from divide [a,c] [km] | Elevation [a] Mean [d] [m] | Min [m] | Max [m] | Slope [a] Median [°] | Q₁-Q₃ [°] | Kurtosis [e] [-] | Total relief [a] [m] | Average local relief [a,f] [m] | Rainfall [d,g] [mm yr⁻¹] | Exposed bedrock without silcrete [h] [%] | Exposed silcrete [h] [%] | Colluvium cover [h] [%] | Gibber cover [h] [%] | Dunes/ aeolian cover [h] [%] | Sand plains [h] [%] | Alluvium cover [h] [%] |
|---|---|---|---|---|---|---|---|---|---|---|---|---|---|---|---|---|---|---|---|
| *NEALES catchment (continued)* | | | | | | | | | | | | | | | | | | | |
| NIL | 192.0 | 1.3 | 2.3 | 368 ± 29 | 281 | 412 | 6 | 3-11 | 2.0 | 131 | 172 | 198 ± 9 | 100 | 0.0 | 0.0 | 0.0 | 0.0 | 0.0 | 0.0 |
| PEA7 | 191.9 | 4287.4 | 212.6 | 245 ± 67 | 97 | 372 | 1 | 1-3 | 13.7 | 275 | 50 | 149 ± 16 | 3.4 | 21.2 | 19.9 | 41.3 | 0.0 | 6.8 | 7.4 |
| NEA3 | 190.7 | 4404.9 | 176.6 | 199 ± 50 | 104 | 351 | 1 | 1-2 | 16.9 | 247 | 37 | 156 ± 6 | 9.6 | 21.0 | 5.4 | 46.8 | 1.0 | 8.8 | 7.4 |
| NEA4 | 91.0 | 0.7 | 2.1 | 83 ± 11 | 63 | 104 | 2 | 1-3 | 1.8 | 41 | 43 | 142 ± 1 | 99.7 | 0.0 | 0.0 | 0.0 | 0.0 | 0.0 | 0.3 |
| PEA8 | 74.7 | 17506.1 | 309.5 | 177 ± 76 | 56 | 418 | 1 | 1-2 | 31.9 | 362 | 35 | 138 ± 13 | 9.6 | 10.1 | 8.3 | 38.5 | 5.6 | 19.3 | 8.6 |
| NEA5 | 0.0 | 27324.4 | 374.8 | 166 ± 74 | 26 | 418 | 1 | 1-2 | 32.7 | 392 | 36 | 142 ± 13 | 10.1 | 10.4 | 6.5 | 43.5 | 4.0 | 16.3 | 9.2 |

a) Based on 1 arc second SRTM DEM.

b) Flow distance to most downstream sampling location derived from watershed delineation in ArcGIS.

c) Flow distance from drainage divide derived from watershed delineation in ArcGIS.

d) Uncertainties expressed at 1-σ level.

e) Kurtosis as indicator for the shape of the slope distribution curve and as measure for representativeness of the mean. High kurtosis values indicate pronounced clustering of slope values around the mean.

f) Catchment average of relief in a 2.5-km radius around every pixel within the catchment.

g) Based on the average of annual mean precipitation rates between the years 1911 and 2000 (Australian Bureau of Meteorology: http://www.bom.gov.au/climate/).

h) Based on 1:1 million surface geology map of Australia (Raymond et al., 2012).

i) Samples from Heimsath et al. (2010); labels H8, 9, 10, 11, 19, and 37 (shown in our Fig. 5B and C) refer to MD-108, -109, -110, -111, -119, -137 in Heimsath et al. (2010).

Table 2: Cosmogenic nuclide data.

| Sample ID | AMS ID (Be/Al) | Latitude[a] [°S] | Longitude[a] [°E] | Material | Mean elevation [m] | Production scaling factor[b] | Sample mass [g qtz] | $^{10}Be/^9Be$ ratio[c,d,e] [10⁻¹⁵] | $^9Be$ carrier mass[f] [mg] | $^{10}Be$ conc.[a] [10³ at g⁻¹] | $^{26}Al/^{27}Al$ ratio[e,g,h,i] [10⁻¹⁵] | $^{27}Al$ ICP conc. [ppm in qtz] | $^{26}Al$ conc.[a] [10³ at g⁻¹] | $^{26}Al/^{10}Be$ ratio[a] |
|---|---|---|---|---|---|---|---|---|---|---|---|---|---|---|
| *FINKE catchment* | | | | | | | | | | | | | | |
| UHugh199 | 147 | -23.811033 | 133.184993 | Fan surface sedi. | 789 | 1.20 | 33.677 | 5091 ± 196[2,C] | 0.459[I] | 4605 ± 185 | - | - | - | - |
| UHugh299 | 148 | -23.811033 | 133.184993 | Fan (0.9 m depth) | 789 | 1.20 | 28.226 | 2201 ± 108[2,C] | 0.374[I] | 1915 ± 99 | - | - | - | - |
| UHugh499 | 150 | -23.809683 | 133.192100 | Fan surface sedi. | 764 | 1.17 | 34.694 | 3515 ± 217[2,C] | 0.370[I] | 2479 ± 157 | - | - | - | - |
| UHugh399 | 149 | -23.809683 | 133.192100 | Fan (2 m depth) | 764 | 1.17 | 29.819 | 802 ± 43[2,C] | 0.457[I] | 781 ± 47 | - | - | - | - |
| B122p | 122P | -23.809683 | 133.192100 | Fan (2.7 m depth) | 764 | 1.17 | 27.519 | 579 ± 44[2,B] | 0.296[I] | 379 ± 32 | - | - | - | - |
| PIO | B6221/a446 | -23.676543 | 132.714092 | Stream sediment | 777 | 1.32 | 40.191 | 897 ± 19[1,A] | 0.294[H] | 495 ± 15 | 1282 ± 37[K] | 97 | 2786 ± 140 | 5.62 ± 0.33 |
| FIN1 | B6222/- | -23.678980 | 132.671712 | Stream sediment | 802 | 1.34 | 30.262 | 607 ± 17[1,A] | 0.297[H] | 450 ± 16 | - | 4211 | - | 5.46 ± 0.33 |
|  | -/a466 | | | | | | 20.240 | - | 0.295[H] | - | 1858 ± 53[L] | 59 | 2454 ± 122 | - |
| FIN2 | B6223/a447 | -23.951370 | 132.774172 | Stream sediment | 761 | 1.30 | 40.916 | 935 ± 18[1,A] | 0.296[H] | 510 ± 15 | 1407 ± 35[K] | 77 | 2412 ± 116 | 4.73 ± 0.27 |
| B123s | 123S | -23.810240 | 133.190935 | Stream sediment | 845 | 1.39 | 21.486 | 438 ± 41[2,B] | 0.268[I] | 322 ± 35 | - | - | - | - |
| ELL | B6227/a454 | -24.087429 | 132.839025 | Stream sediment | 710 | 1.26 | 40.460 | 833 ± 25[1,A] | 0.297[H] | 461 ± 17 | 1317 ± 34[K] | 80 | 2358 ± 114 | 5.11 ± 0.31 |
| FIN3 | B6224/a451 | -24.552860 | 133.238430 | Stream sediment | 691 | 1.24 | 40.369 | 987 ± 20[1,A] | 0.297[H] | 548 ± 16 | 1407 ± 36[K] | 87 | 2744 ± 132 | 5.01 ± 0.28 |
| PAL | B6228/a455 | -24.750439 | 133.186722 | Stream sediment | 638 | 1.20 | 35.035 | 945 ± 16[1,A] | 0.298[H] | 606 ± 17 | 1543 ± 39[K] | 91 | 3149 ± 151 | 5.20 ± 0.29 |
| FIN4 | B6225/a452 | -24.929894 | 133.640178 | Stream sediment | 617 | 1.18 | 40.230 | 1061 ± 19[1,A] | 0.297[H] | 590 ± 17 | 1451 ± 36[K] | 87 | 2813 ± 135 | 4.77 ± 0.27 |
| HUG | B6229/a456 | -24.677768 | 134.059998 | Stream sediment | 573 | 1.14 | 40.163 | 1073 ± 20[1,A] | 0.297[H] | 598 ± 17 | 1381 ± 35[K] | 86 | 2656 ± 128 | 4.44 ± 0.25 |
| FIN5 | B6226/a453 | -25.217346 | 134.241625 | Stream sediment | 576 | 1.15 | 40.245 | 1045 ± 17[1,A] | 0.298[H] | 582 ± 16 | 1281 ± 35[K] | 88 | 2531 ± 124 | 4.34 ± 0.25 |
| S05/04 | - | -25.679883 | 134.854368 | Stream sediment | 539 | 1.12 | - | - | - | 541 ± 16 | - | - | 2763 ± 187 | 5.10 ± 0.38 |
| *MACUMBA catchment* | | | | | | | | | | | | | | |
| COO | B5947/A2680 | -27.162479 | 134.375555 | Stream sediment | 270 | 0.97 | 40.128 | 2944 ± 32[1,D] | 0.317[J] | 1695 ± 42 | 5259 ± 258 *[M] | 59 | 6868 ± 481 | 4.05 ± 0.30 |
| ALB3 | B6041/A2782 | -27.129882 | 134.389281 | Stream sediment | 268 | 0.97 | 41.829 | 2596 ± 36[1,E] | 0.305[H] | 1427 ± 38 | 4971 ± 121 *[N] | 57 | 6348 ± 353 | 4.45 ± 0.27 |
| ALB2 | B6040/A2781 | -27.130915 | 134.434604 | Stream sediment | 289 | 0.99 | 40.291 | 2463 ± 24[1,E] | 0.305[H] | 1404 ± 34 | 4147 ± 112 *[N] | 62 | 5697 ± 324 | 4.06 ± 0.25 |
| OLA[i] | B6038/A2779 | -27.164221 | 134.621190 | Stream sediment | 268 | 0.97 | 40.504 | 7470 ± 28[1,E] | 0.302[H] | 4200 ± 95 | 1099 ± 55[N] | 252 | 6183 ± 439 | 1.47 ± 0.11 |
| ALB1 | B6039/A2780 | -27.153811 | 134.753684 | Stream sediment | 418 | 1.08 | 40.322 | 2343 ± 17[1,E] | 0.305[H] | 1335 ± 31 | 2613 ± 122 *[N] | 96 | 5584 ± 383 | 4.18 ± 0.30 |
| MAC | B5708/A2588 | -27.197277 | 135.716094 | Stream sediment | 322 | 1.00 | 40.354 | 2774 ± 23[1,F] | 0.322[J] | 1612 ± 38 | 2279 ± 188 *[O] | 95 | 4838 ± 467 | 3.00 ± 0.30 |
| *NEALES catchment* | | | | | | | | | | | | | | |
| PEA-BR2 | B6026/A2734 | -27.960354 | 134.199993 | Bedrock | 252 | 0.97 | 13.099 | 358 ± 7[1,G] | 0.295[H] | 609 ± 18 | 1327 ± 65[L] | 126 | 3745 ± 262 | 6.15 ± 0.47 |
| PEA-BR3[i] | B6028/A2736 | -27.945442 | 134.392228 | Bedrock | 255 | 0.98 | 16.723 | 9898 ± 34[1,G] | 0.294[H] | 13126 ± 296 | 3932 ± 123[L] | 93 | 8128 ± 479 | 0.62 ± 0.04 |
| PEA-BR4 | B6027/A2735 | -28.199020 | 134.775937 | Bedrock | 219 | 0.95 | 17.326 | 1302 ± 12[1,G] | 0.294[H] | 1670 ± 40 | 6623 ± 239[L] | 67 | 9977 ± 615 | 5.97 ± 0.40 |
| PEA1 | B5703/A2583 | -27.348124 | 133.969076 | Stream sediment | 355 | 1.04 | 40.155 | 5386 ± 34[1,F] | 0.318[J] | 3105 ± 72 | 7618 ± 306 *[O] | 66 | 11292 ± 724 | 3.64 ± 0.25 |
| PEA2 | B5704/A2584 | -27.943413 | 134.153153 | Stream sediment | 281 | 1.00 | 40.201 | 7236 ± 85[1,F] | 0.318[J] | 4172 ± 105 | 9782 ± 330 *[O] | 73 | 15885 ± 958 | 3.81 ± 0.25 |
| PEA4 | B6034/A2775 | -28.210212 | 134.481050 | Stream sediment | 259 | 0.99 | 41.209 | 7665 ± 39[1,E] | 0.303[H] | 4250 ± 98 | 9098 ± 181 *[N] | 71 | 14471 ± 779 | 3.41 ± 0.20 |

*Continued on next page.*

| Sample ID | AMS ID (Be/Al) | Latitude[a] [°S] | Longitude[a] [°E] | Material | Mean elevation [m] | Production scaling factor[b] | Sample mass [g qtz] | $^{10}Be/^{9}Be$ ratio[c,d,e] [$10^{-15}$] | $^{9}Be$ carrier mass[f] [mg] | $^{10}Be$ conc.[a] [$10^3$ at g$^{-1}$] | $^{26}Al/^{27}Al$ ratio[e,g,h,i] [$10^{-15}$] | $^{27}Al$ ICP conc. [ppm in qtz] | $^{26}Al$ conc.[a] [$10^3$ at g$^{-1}$] | $^{26}Al/^{10}Be$ ratio[a] |
|---|---|---|---|---|---|---|---|---|---|---|---|---|---|---|
| **NEALES catchment (continued)** | | | | | | | | | | | | | | |
| PEA5 | B5705/A2585 | -28.203679 | 134.665591 | Stream sediment | 248 | 0.97 | 40.376 | 5656 ± 44 [I,F] | 0.320 [J] | 3261 ± 77 | 7080 ± 348 *[O] | 70 | 11006 ± 772 | 3.38 ± 0.25 |
| NEA1 | B5948/A2681 | -27.393263 | 135.263533 | Stream sediment | 207 | 0.93 | 40.135 | 1978 ± 33 [I,D] | 0.310 [J] | 1111 ± 31 | 3099 ± 235 *[M] | 64 | 4460 ± 405 | 4.02 ± 0.38 |
| PEA6 | B6035/A2776 | -28.313134 | 134.946048 | Stream sediment | 226 | 0.96 | 40.117 | 5460 ± 22 [I,E] | 0.305 [J] | 3134 ± 71 | 6174 ± 232 *[N] | 75 | 10287 ± 643 | 3.28 ± 0.22 |
| NEA2 | B6036/A2777 | -27.867062 | 135.123488 | Stream sediment | 187 | 0.92 | 40.093 | 1220 ± 13 [I,E] | 0.305 [J] | 700 ± 17 | 1239 ± 74 *[N] | 119 | 3296 ± 258 | 4.71 ± 0.39 |
| NIL | B5709/- | -28.482968 | 135.999887 | Stream sediment | 368 | 1.08 | 40.187 | 848 ± 27 [I,F] | 0.322 [J] | 496 ± 19 | - | 135 | - | 6.06 ± 0.38 |
|  | -/a464 |  |  |  |  |  | 17.609 |  | 0.295 [H] |  | 955 ± 27 [L] | 141 | 3005 ± 149 |  |
| PEA7 | B6032/A2740 | -28.11550 | 135.082709 | Stream sediment | 245 | 0.97 | 40.531 | 2789 ± 27 [I,G] | 0.294 [H] | 1523 ± 37 | 4083 ± 118 *[N] | 62 | 5662 ± 327 | 3.72 ± 0.23 |
| NEA3 | B6037/A2778 | -27.620241 | 135.427262 | Stream sediment | 199 | 0.93 | 40.274 | 2188 ± 14 [I,E] | 0.304 [H] | 1246 ± 29 | 2733 ± 90 *[N] | 71 | 4307 ± 258 | 3.46 ± 0.22 |
| NEA4 | B6031/- | -27.900861 | 135.802884 | Stream sediment | 83 | 0.85 | 40.488 | 516 ± 6 [I,G] | 0.293 [H] | 282 ± 7 | - | 124 | - | 8.86 ± 0.34 |
|  | -/a467 |  |  |  |  |  | 20.075 |  | 0.301 [H] |  | 597 ± 20 *[L] | 124 | 1650 ± 87 |  |
| PEA8 | B5706/- | -28.035828 | 135.797000 | Stream sediment | 177 | 0.92 | 40.365 | 1383 ± 17 [I,F] | 0.320 [J] | 799 ± 20 | - | 98 | - | 4.60 ± 0.25 |
|  | -/a462 |  |  |  |  |  | 16.504 |  | 0.299 [H] |  | 1715 ± 46 [L] | 96 | 3671 ± 179 |  |
| NEA5 | B5707/- | -28.114007 | 136.300039 | Stream sediment | 166 | 0.91 | 40.231 | 1329 ± 16 [I,F] | 0.322 [H] | 774 ± 20 | - | 112 | - | 4.39 ± 0.24 |
|  | -/a463 |  |  |  |  |  | 20.075 |  | 0.296 [H] |  | 1363 ± 35 [L] | 112 | 3400 ± 164 |  |

a) Coordinates indicate the location of the catchment outlet on the 30 m SRTM DEM; values referenced to WGS84 Datum.

b) Combined atmospheric pressure/latitude scaling factor following the time-independent scaling scheme of Stone (2000).

c) $^{10}Be/^{9}Be$ ratios were normalised to standards 1) SRM KN-5-2 (nominal ratio of $8.558 \times 10^{-15}$; 2% reproducibility error) (Nishiizumi et al., 2007), and 2) NIST4325 (nominal ratio $27{,}900 \times 10^{-15}$; 3% reproducibility error).

d) Corrected for batch procedural blanks of: A) $1.69 \pm 0.92 \times 10^{-15}$, B) $51.28 \pm 7.99 \times 10^{-15}$, C) $39.26 \pm 12.47 \times 10^{-15}$, D) $7.83 \pm 2.10 \times 10^{-15}$, E) $5.50 \pm 0.70 \times 10^{-15}$, F) $2.94 \pm 0.74 \times 10^{-15}$, G) $6.24 \pm 0.95 \times 10^{-15}$.

e) Uncertainties expressed at 1-σ level.

f) Concentrations of $^{9}Be$ carrier solutions are: H) 1090 ± 15 ppm, I) unknown, J) 1128 ± 22 ppm.

g) $^{26}Al/^{27}Al$ ratios marked with * were blank-corrected using the respective blank's $^{26}Al$ count rate.

h) $^{26}Al/^{27}Al$ Al were normalised to SRM KN-4-2 with a nominal ratio of $30{,}960 \times 10^{-15}$ (Nishiizumi, 2004).

i) Corrected for batch procedural blanks of: K) $4.33 \pm 1.53 \times 10^{-15}$, L) $13.57 \pm 2.36 \times 10^{-15}$, M) $10.36 \pm 3.76 \times 10^{-15}$, N) $22.06 \pm 5.35 \times 10^{-15}$, O) $321.34 \pm 25.44 \times 10^{-15}$.

j) Samples were excluded from further analyses since $^{10}Be$ abundances are unnaturally high.

## 4  Results

All catchments display low slope gradients overall $\leq$1–3°, although steeper slopes are rather more common in the Finke (Table 1). Many catchments exhibit a substantial proportion (>50%) of bedrock outcrop, especially in the northern Finke strike-ridge country, in the silcrete-tablelands in the western Macumba and Neales, and in the Peake and Denison Range in
the lower Neales catchment. Elsewhere the landscape is draped with a largely continuous cover of stony soil mantles, alluvial plains, and aeolian deposits in varying proportions (Table 1). We use 'fraction of bedrock and colluvium' in scatter plots to represent the proportion of source-area terrain upstream of our stream samples (Figs. 4 and 5)—in other words, the area producing the source-area signal that we track downstream through the sediment-routing system.

### 4.1  $^{10}$Be abundances in sediment

$^{10}$Be abundances in stream sediment span 0.3 to 4.3 x $10^6$ atoms g$^{-1}$ and vary widely between subcatchments (Table 2). Large drainage areas and down-system samples consistently yield $^{10}$Be levels at the low end of the range, whereas smaller headwater streams are more variable and tend to span the full range (Fig. 4A). Similarly, relatively low $^{10}$Be levels generally follow areas with >100 m mean relief (almost exclusively within the Finke catchment) and areas of lower relief yield a wide range (Fig. 4B). No relationship exists between $^{10}$Be and fraction of bedrock and colluvium in the Finke and Macumba, but high $^{10}$Be among
the five rocky headwaters of the Peake subcatchment decreases downstream as sediment-cover expands (Fig. 4C). These small streams draining the silcrete mesas of the Peake (Fig. 2) yield the highest $^{10}$Be levels in stream sediment (Fig. 4). Conversely, the lower Peake receives sediment from the locally steep Peake and Denison Range whose small headwater streams yield some of the lowest $^{10}$Be in our dataset (Figs. 2 and 4). The effect of such inputs is seen in the low $^{10}$Be from the lower Neales samples PEA8 and NEA5 (Figs. 2 and 5H).

### 4.2  Modelled denudation rates and apparent burial ages in sediment

Overall $^{26}$Al/$^{10}$Be ratios in sediment span 1.5–6.1, with the majority $\sim$3–5 (20 samples) (Table 2). The Finke displays generally higher $^{26}$Al/$^{10}$Be ratios (4.7–5.2, interquartile range) relative to the Macumba and Neales (3.5–4.4). Deviation from the steady-state erosion island is typically attributed to one or more episodes of burial-exposure, yet it has been long understood that particle burial cannot be differentiated from non-steady exhumation based on the $^{26}$Al/$^{10}$Be ratio (Gosse and Phillips, 2001).
Hence, we emphasise that our modelled apparent burial ages (Table 3) serve primarily as a measure of deviation from the steady-state erosion curve (Fig. 6). For most of our samples (n = 21) deviations cluster between $\sim$400 and 800 k.y. and range up to $\sim$1.1 m.y. (Table 3). Low deviations <400 k.y. are exclusively observed in small headwater streams (PIO, FIN1, NEA4, NIL, PEA2), although deviations close to the erosion island are difficult to discriminate due to the spread of uncertainties — the erosion island itself does not accommodate uncertainties in production rate.

30       Assuming that sediment samples have been continuously exposed at the surface, without decay of nuclides due to burial, the $^{10}$Be abundances yield slow catchment-scale denudation rates between 0.3 and 11.0 m m.y.$^{-1}$ (Table 3). When corrected for the 'apparent burial age', as calculated above, denudation rates lower slightly to 0.2–8.1 m m.y.$^{-1}$ (Table 3).

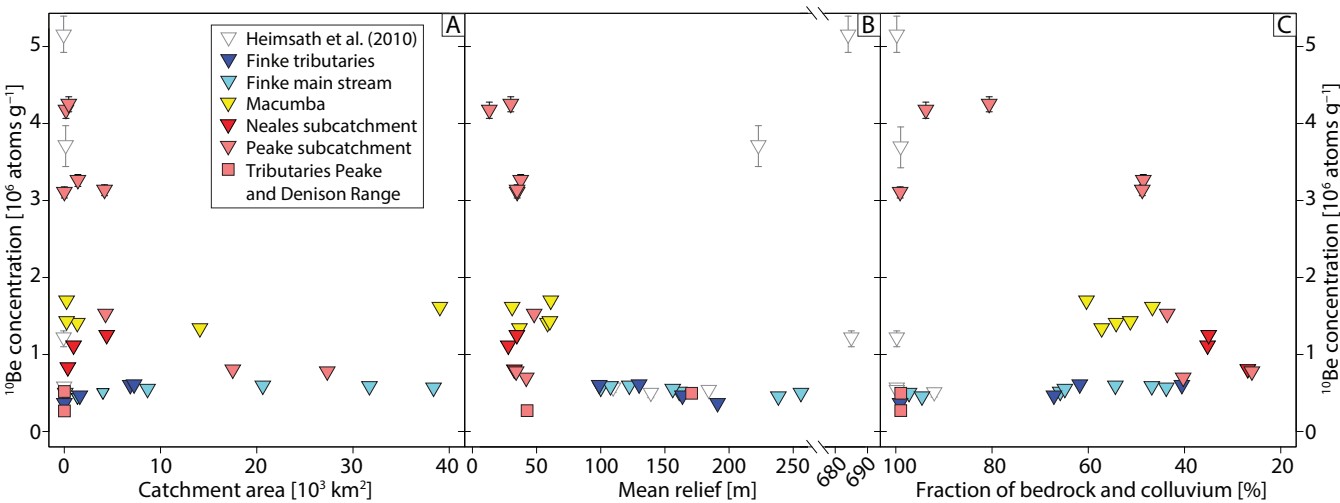

**Figure 4.** $^{10}$Be abundances (normalised to sea-level high-latitude) measured in stream sediment relative to A) drainage area, B) mean relief, C) fraction of exposed bedrock and colluvium cover. Finke samples are blue and white triangles (light blue – trunk stream; dark blue and white – tributaries), Macumba samples are yellow triangles, and Neales samples are red triangles and squares (dark – Neales subcatchment, light – Peake subcatchment, squares – Peake and Denison Range).

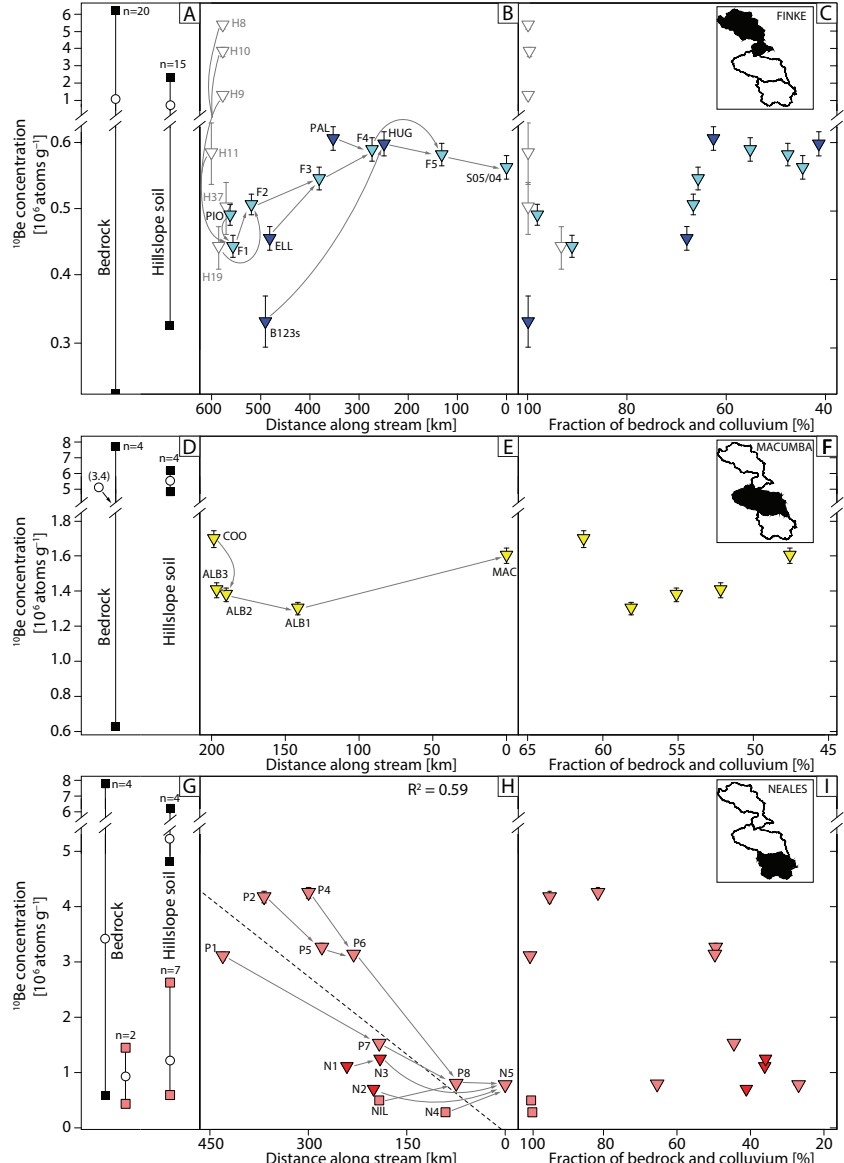

**Figure 5.** $^{10}$Be abundances of bedrock and stream sediment from the Finke (panels A, B, C) showing trunk streams (light-blue triangles) and tributaries (dark-blue and white triangles), Macumba (D, E, F), and Neales (G, H, I) rivers. The Neales data are further subdivided into the subcatchments of Peake (light-red triangles), Neales (dark-red triangles), and Peake and Denison Range (light-red squares). Panels A, D, and G show $^{10}$Be abundances in bedrock and hillslope soil as median (open circles) and full range (black squares for MacDonnell Ranges and silcrete, and light-red squares for Peake and Denison Range). Panels B, E, and H show $^{10}$Be abundances in stream sediment relative to the distance along-stream from most downstream sample—note that we have reversed the x-axes in all panels to illustrate our data from source-to-sink, left to right. Arrows indicate stream trajectories (sample labels corresponding to Tables: F1-5 are FIN1-5, N1-5 are NEA1-5, and P1-8 are PEA1-8; H denotes samples from Heimsath et al., 2010). Panels C, F, and I, show the fraction of exposed bedrock and colluvium cover. Note that previously published data are included in panel A (Struck et al., 2018; Heimsath et al., 2010) and panels D and G (Struck et al., 2018; Fujioka et al., 2005)(see Table A3). All nuclide data are normalised to sea-level high-latitude.

Table 3: Basin-wide erosion rates and apparent burial ages.

| Sample ID | Surface erosion rate[a,b] [m m.y.$^{-1}$] | Apparent burial age[c,d] [k.y.] | Surface erosion rate accounted for burial[c,d] [m m.y.$^{-1}$] |
|---|---|---|---|
| *FINKE catchment* | | | |
| PIO | $7.46 \pm 0.25$ | $266^{+152}_{-88}$ | $6.45^{+0.80}_{-0.60}$ |
| FIN1 | $8.41 \pm 0.32$ | $340^{+100}_{-113}$ | $7.02^{+1.33}_{-0.53}$ |
| FIN2 | $7.14 \pm 0.23$ | $607^{+152}_{-91}$ | $5.12^{+0.60}_{-0.50}$ |
| B123s | $10.96 \pm 1.19$ | - | - |
| ELL | $7.69 \pm 0.31$ | $465^{+154}_{-103}$ | $5.97^{+0.91}_{-0.65}$ |
| FIN3 | $6.31 \pm 0.21$ | $475^{+152}_{-94}$ | $4.85^{+0.58}_{-0.47}$ |
| PAL | $5.47 \pm 0.17$ | $399^{+139}_{-96}$ | $4.37^{+0.54}_{-0.37}$ |
| FIN4 | $5.54 \pm 0.18$ | $566^{+135}_{-95}$ | $4.03^{+0.51}_{-0.39}$ |
| HUG | $5.27 \pm 0.17$ | $685^{+149}_{-94}$ | $3.59^{+0.45}_{-0.34}$ |
| FIN5 | $5.45 \pm 0.17$ | $743^{+139}_{-89}$ | $3.59^{+0.40}_{-0.32}$ |
| S05/04 | $5.52 \pm 0.18$ | $505^{+200}_{-126}$ | $4.18^{+0.64}_{-0.48}$ |
| *MACUMBA catchment* | | | |
| COO | $1.28 \pm 0.04$ | $568^{+170}_{-101}$ | $0.87^{+0.13}_{-0.11}$ |
| ALB3 | $1.59 \pm 0.05$ | $471^{+153}_{-101}$ | $1.17^{+0.18}_{-0.12}$ |
| ALB2 | $1.66 \pm 0.05$ | $638^{+140}_{-86}$ | $1.10^{+0.14}_{-0.11}$ |
| ALB1 | $1.95 \pm 0.06$ | $625^{+185}_{-107}$ | $1.32^{+0.18}_{-0.15}$ |
| MAC | $1.42 \pm 0.04$ | $1115^{+242}_{-126}$ | $0.66^{+0.13}_{-0.11}$ |
| *NEALES catchment* | | | |
| PEA-BR2 | $4.41 \pm 0.15$ | $28^{+115}_{-14}$ | $4.34^{+0.20}_{-0.37}$ |
| PEA-BR4 | $1.23 \pm 0.04$ | $0^{+69}_{-0}$ | $1.22^{+0.05}_{-0.07}$ |
| PEA1 | $0.60 \pm 0.02$ | $532^{+144}_{-85}$ | $0.38^{+0.06}_{-0.05}$ |
| PEA2 | $0.33 \pm 0.02$ | $295^{+117}_{-82}$ | $0.24^{+0.05}_{-0.04}$ |
| PEA4 | $0.31 \pm 0.01$ | $454^{+116}_{-76}$ | $0.18^{+0.04}_{-0.03}$ |
| PEA5 | $0.50 \pm 0.02$ | $592^{+150}_{-84}$ | $0.28^{+0.05}_{-0.05}$ |
| NEA1 | $2.07 \pm 0.07$ | $719^{+240}_{-137}$ | $1.32^{+0.24}_{-0.19}$ |
| PEA6 | $0.52 \pm 0.02$ | $650^{+143}_{-80}$ | $0.28^{+0.05}_{-0.04}$ |
| NEA2 | $3.55 \pm 0.10$ | $526^{+203}_{-127}$ | $2.61^{+0.40}_{-0.31}$ |
| NIL | $6.11 \pm 0.26$ | $30^{+5}_{-10}$ | $6.16^{+0.31}_{-0.21}$ |
| PEA7 | $1.46 \pm 0.05$ | $758^{+159}_{-94}$ | $0.88^{+0.12}_{-0.10}$ |
| NEA3 | $1.79 \pm 0.05$ | $934^{+161}_{-89}$ | $0.98^{+0.12}_{-0.11}$ |
| NEA4 | $9.07 \pm 0.25$ | $188^{+123}_{-63}$ | $8.13^{+0.82}_{-0.62}$ |
| PEA8 | $3.04 \pm 0.09$ | $542^{+137}_{-89}$ | $2.20^{+0.26}_{-0.20}$ |
| NEA5 | $3.11 \pm 0.09$ | $633^{+134}_{-87}$ | $2.13^{+0.24}_{-0.19}$ |

a) Calculated from $^{10}$Be concentrations with the single-nuclide-erosion tool of CosmoCalc 3.0 (Vermeesch, 2007), using the time-independent scaling scheme of Stone (2000) and production mechanisms based on Granger and Muzikar (2001).

b) Uncertainties expressed at 1-σ level.

c) Calculated using the CosmoCalc 3.0 (Vermeesch, 2007) burial-erosion tool. The calculation assumes a simple burial scenario, namely, one episode of erosion followed by one episode of burial. The calculation does not account for post-burial re-exposure.

d) Uncertainties expressed at 1 standard deviation (i.e., 68[th] percentile).

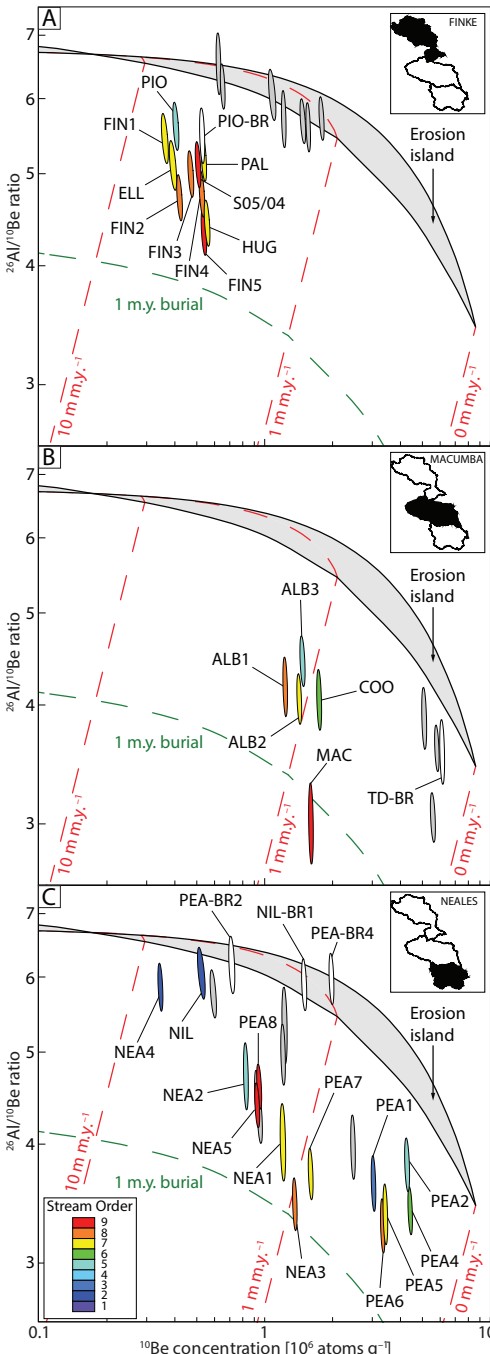

**Figure 6.** Two-nuclide logarithmic plots showing $^{26}$Al/$^{10}$Be ratios (normalised to sea-level high-latitude) in bedrock (white ellipses, 'BR'), hillslope soil (grey ellipses), and stream sediments (colour-coded by stream order; low – blue, high – red), A) Finke catchment, B) Macumba catchment, C) Neales catchment. Grey areas represent simple exposure/erosion history (erosion island). Shown are erosion rates (red dashes) and 1 m.y. burial isochrons (green). Continuously exposed samples should plot within the steady-state erosion island; samples plotting left of the erosion island indicate a history of post-exhumation burial(s) and/or non-steady exhumation.

# 5 Down-system variation of $^{10}$Be-$^{26}$Al in the western Eyre Basin

## 5.1 Lithology and the $^{10}$Be-$^{26}$Al source-area signal

$^{10}$Be levels measured in source-area bedrock and hillslope soil vary widely between our three catchments, but broadly concur within each catchment as reported by Struck et al. (2018) and shown for comparison with samples from the stream network in

Figure 5. Lithology is primarily responsible for the wide variation in erosion rates measured on bedrock surfaces in the western Eyre Basin in the order (from slowest to fastest): silcrete, quartzite, sandstone, conglomerate (cf. Fig. 13 in Struck et al., 2018). Compiling bedrock erosion-rate data (n = 26) from Fujioka (2007); Heimsath et al. (2010), and Struck et al. (2018) yields interquartile ranges of 0.2–4.4 m m.y.$^{-1}$ (n = 4) on silcrete mesas in the Oodnadatta Tablelands; 1.6–4.8 m m.y.$^{-1}$ (n = 15) on quartzite-sandstone ridges in the Western MacDonnell Ranges; 1.8–7.3 m m.y.$^{-1}$ (n = 2) on quartzite-sandstone in the Peake

and Denison Range; and 6.7–6.8 m m.y.$^{-1}$ (n = 5) on conglomerate in the Western MacDonnell Ranges. These differences in source-area erosion rates are also reflected in the $^{10}$Be levels measured in stream sediments downstream (Fig. 4A), which translate to catchment erosion rates (interquartile ranges) of 4.1–5.8 m m.y.$^{-1}$ in the Finke, 0.9–1.2 m m.y.$^{-1}$ in the Macumba, and 0.3–2.2 m m.y.$^{-1}$ in the Neales. The western headwaters of the Peake yield 0.2–0.4 m m.y.$^{-1}$, which are among the slowest catchment-scale erosion rates ever measured (Table 3).

Our bedrock samples overall have experienced a history of continuous surface exposure or deviate slightly from the steady-state condition (Fig. 6A,C). As proposed by Struck et al. (2018), the minor deviation from the steady-state erosion curve (Fig. 6A) may be the result of non-steady exhumation—termed 'two-speed exhumation'. Considering the very low erosion rates (<1 m m.y.$^{-1}$) we report for the western Eyre Basin, $^{26}$Al/$^{10}$Be ratios will decrease (<6.75) throughout the rock column owing to the faster decay of $^{26}$Al relative to $^{10}$Be. Under these conditions a sudden pulse of erosion due to recent soil-stripping,

for instance, will cause surface sample $^{26}$Al/$^{10}$Be ratios to deviate from the steady-state erosion curve (Fig. 6). Two-speed exhumation provides a viable alternative to cyclic exposure-burial that is usually invoked to account for low $^{26}$Al/$^{10}$Be ratios (Struck et al., 2018).

## 5.2 $^{10}$Be-$^{26}$Al in the Finke sediment-routing system

The prominent strike ridges and hillslope soil mantles of the MacDonnell Ranges (Fig. 3A) contain a wide range of abundances

of $^{10}$Be $\sim$0.2–6.5 x 10$^6$ atoms g$^{-1}$ (Fig. 5A) which appears to be driven by bedrock lithology (cf. Fig. 13 in Struck et al., 2018). In some cases, small alluvial fans form intermediate storages of sediment prior to it entering the stream network, but more commonly bedrock ridges feed sediment directly to low-order headwater streams (Fig. 5B). High $^{10}$Be (1–5 x 10$^6$ atoms g$^{-1}$) occurs in streams draining resistant quartzite ridges, whereas streams from sandstone-siltstone ridges and low conglomerate hills yield $\sim$0.3–0.6 x 10$^6$ atoms g$^{-1}$. From the headwaters $^{10}$Be increases slightly over $\sim$300 km downstream (Fig. 5B) to

where the channel and floodplain system broadens to unconfined alluvial plains and dune fields (at FIN4, Fig.2) and from here remains constant downstream. This slight rise in $^{10}$Be downstream coincides with the shrinking fraction of bedrock and colluvium (Fig. 5C) and rise in the extent of sediment cover.

The bedrock and soil samples contain a minor burial signal (<0.3 m.y.) (Fig. 7A), which is transmitted to sediments of the headwater streams (Fig. 7B). Similar to the down-system trends in $^{10}$Be, the burial signal increases downstream over ~450 km then remains constant (or decreases slightly) to the most downstream sample (Fig. 7B); the apparent burial signal also shows a convincing negative correlation ($R^2$ = 0.68) with the fraction of bedrock and colluvium (Fig. 7C).

## 5.3    $^{10}$Be-$^{26}$Al in the Macumba-Neales sediment-routing system

The Macumba and Neales river catchments both drain the silcrete-mesa country of the Oodnadatta Tablelands, which means that their sediment-routing systems share key physiographic and lithological controls. We plot their stream sediment data separately in Figures 5 and 7, but the bedrock and soil data (Figs. 5D,G and 7D,G) are treated as regionally representative of the Oodnadatta Tablelands.

Silcrete duricrust forms a caprock that is exceptionally resistant to weathering (Struck et al., 2018) and hence the mesa surfaces tend to accumulate very high $^{10}$Be abundances. Based on their work in the Negev, Boroda et al. (2014) propose that the erosion rate of caprock-mesas scales with their size and extent. Parallel slope retreat, with negligible vertical erosion, predominates on wide tableland plateaus and with ongoing mesa reduction the rate of vertical and horizontal erosion increases to a maximum at the tor-stage. Our four samples from silcrete mesas in the Neales and Macumba catchments are intended to represent the full range of bedrock erosion rates ($^{10}$Be abundances)—starting with a slowly eroding broad plateau (TD-BR ~5.2–7.7 x $10^6$ atoms g$^{-1}$) to a dissected mesa (PEA-BR4 ~1.7 x $10^6$ atoms g$^{-1}$) and finally a tor (PEA-BR2 ~0.6 x $10^6$ atoms g$^{-1}$). The western headwaters of the Neales and Peake subcatchments dissect the eastern edge of a continuous silcrete caprock plateau (Fig. 2). Given that the degree of mesa dissection increases in the down-system direction (west-to-east), according to Boroda et al. (2014), we can predict that $^{10}$Be supply to the stream network decreases downstream—and this is essentially what we find. Extremely high to rather low $^{10}$Be content of mesa bedrock overlaps with data from hillslope soil mantles (Fig. 5G), and the high $^{10}$Be accumulated on the flat, undissected silcrete plateau is transmitted into the westernmost headwater streams of the Peake subcatchment (Fig. 5H). In contrast, the far more dissected areas drained by the Neales and Macumba headwater streams yield relatively low $^{10}$Be (Fig. 5E,H). From the headwaters of the Peake $^{10}$Be decreases sharply over ~200–250 km to levels matching the Neales and Macumba streams (Fig. 5H), which both show limited variation over ~200 km downstream (Fig. 5E,H). These downstream trends are broadly accompanied by the reduction in bedrock and expansion of sediment cover (Fig. 5H). The Peake and Denison Range in the southeast corner of the Neales catchment (Fig. 2) exerts an important effect on the sediment-routing system. Samples from quartzite-sandstone bedrock together with soil (Fig. 5G) demonstrate that the high-relief and weaker lithology is driving erosion rates that are much faster relative to the Oodnadatta Tablelands to the west. Stream sediments from these ranges enter the lower reaches of the Peake and Neales rivers where they notably depress $^{10}$Be abundances (Fig. 5H).

The burial signal measured in bedrock and hillslope soil mantles (<0.6 m.y.) is transmitted into headwater streams with fairly similar (or slightly increased) apparent burial ages (Fig. 7D,G). A potential source of low $^{26}$Al/$^{10}$Be material is generated by fluvial gully-heads that undermine the caprock, yielding deeply shielded (>3 m) material from beneath the silcrete. The Macumba undergoes a notable increase in burial signal over ~140 km downstream (Fig. 7E), whereas the Neales and Peake

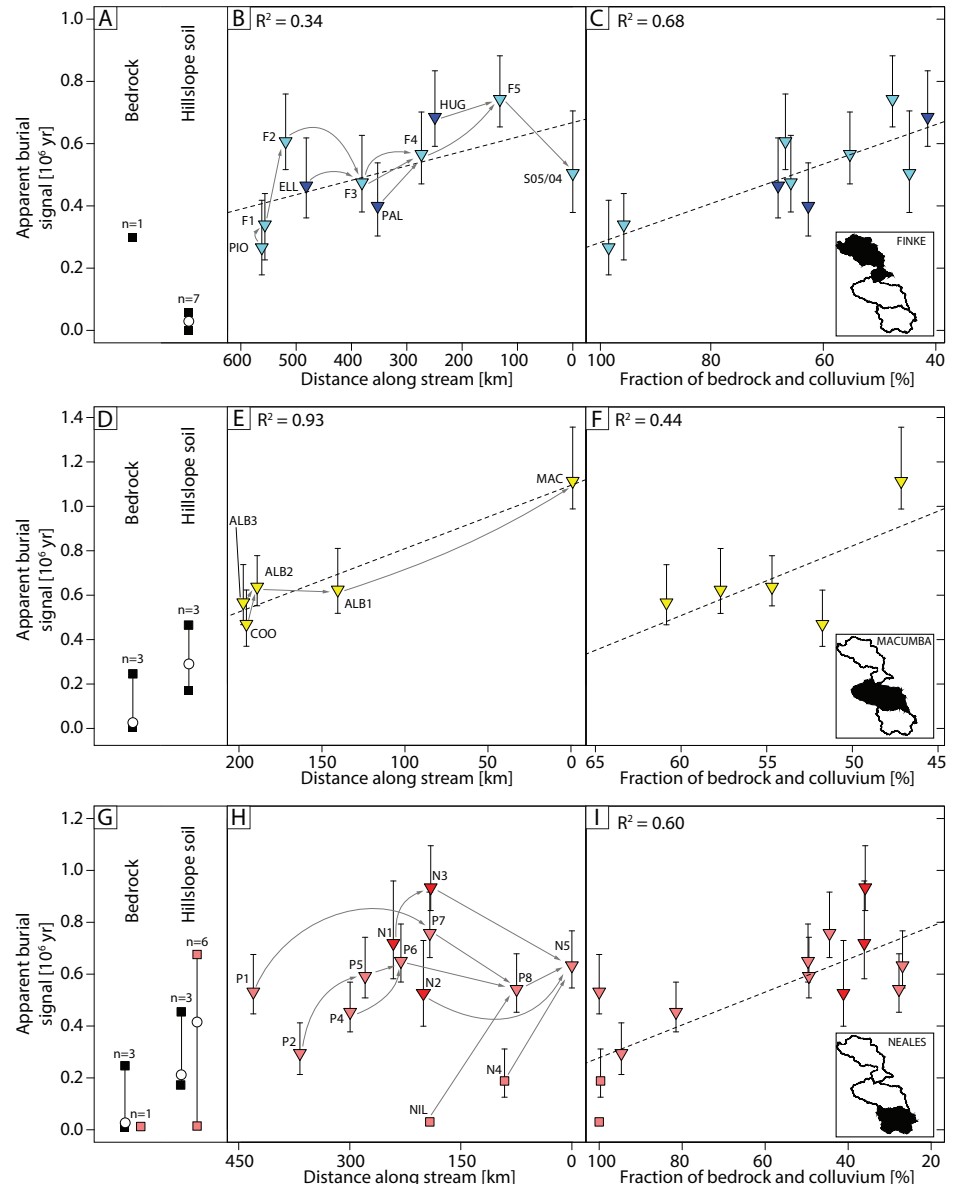

**Figure 7.** Apparent burial ages of bedrock and stream sediment from the Finke (panels A, B, C) showing trunk streams (light-blue triangles) and tributaries (dark-blue and white triangles), Macumba (D, E, F), and Neales (G, H, I) rivers. The Neales data are further subdivided into the subcatchments of Peake (light-red triangles), Neales (dark-red triangles), and Peake and Denison Range (light-red squares). Panels A, D, and G show apparent burial ages in bedrock and hillslope soil as median (open circles) and full range (black squares for MacDonnell Ranges and silcrete, and light-red squares for Peake and Denison Range). Panels B, E, and H show apparent burial ages in stream sediment relative to the distance along-stream from most downstream sample—note that we have reversed the x-axes in all panels to illustrate our data from source-to-sink, left to right. Arrows indicate stream trajectories (sample labels corresponding to Tables: F1-5 are FIN1-5, N1-5 are NEA1-5, and P1-8 are PEA1-8). Panels C, F, and I, show the fraction of exposed bedrock and colluvium cover.

subcatchments show a slight increase in burial over ~200 km until this trend is disrupted by inputs from the Peake and Denison Range (Fig. 7H). Both the Macumba and Neales networks show a broad increase in burial signal relative to the fraction of sediment cover (Fig. 7F,I).

# 6    Factors that modify the $^{10}$Be-$^{26}$Al source-area signal

Cosmogenic nuclide inventories in sediment can be modified in the sediment-routing system via: i) inputs from faster eroding areas, or ii) particles with notably longer exposure histories, including particles buried in transit. We have evidence of the first case where sediment yield from the faster-eroding Peake and Denison Range (Fig. 2) dilutes the high $^{10}$Be and depresses the burial signal emanating from the Peake and Neales subcatchments (Figs. 5 and 7). However, the main modification to the $^{10}$Be-$^{26}$Al source-area inventory appears to be the downstream increase in the burial signal (Fig. 7). This modification indicates

that samples downstream incorporate a growing fraction of particles derived from temporary storage. Such particles are likely to be a mix of those that have acquired additional nuclides during near-surface (<1–2 m) exposure to secondary cosmic rays plus those more deeply buried (i.e., >2–3 m). Only burial can slow down nuclide production, but deep-burial is not essential for lowering $^{26}$Al/$^{10}$Be—even shallow burial can cause deviation from the steady-state erosion curve over timescales on the same order as the $^{26}$Al half-life ~0.7 m.y. (cf. Fig. 14 in Struck et al., 2018). The correlation shown between burial signal

and increasing sediment cover (Figs. 7 and 8) is presumably the result of samples assimilating input from storages with long exposure histories that include some (possibly deep) burial. We identify four key sources for such material: i) alluvial fans, ii) desert pavements, iii) floodplains and palaeo-alluvial plains, and iv) aeolian dunes. Together these landforms span >50% of the total catchment area in the lower stream reaches (Figs. 4 and 7; Table 1).

Alluvial fans are intermediate storages at the transition from hillslopes to the fluvial network, hence they may provide the

first opportunity for alteration of the source-area signal. Cosmogenic nuclide depth-profiles measured in two typical fans of the upper Finke yield depositional ages of 188–289 k.y. (Struck et al., 2018) and ~438 to 1474 k.y. (Fig. A1). If this is representative of alluvial fans in the region, then we can suggest that alluvial fans play an important role in burial signal development for particles entering headwater streams. Sometimes observed mantling older fans, desert pavement (gibber) occurs throughout the sediment-routing system and nuclide-derived residence times of $10^5$–$10^6$ yr demonstrate its extreme

longevity (Fujioka et al., 2005; Fisher et al., 2014; Struck et al., 2018). Gibbers break off and disperse directly from bedrock outcrop, or they form at the bedrock-soil interface and rise to the surface over time—a process that imparts very low $^{26}$Al/$^{10}$Be ratios (Struck et al., 2018). Such gibbers released into streams, together with the underlying aeolian soils held in long-term shallow-burial, are likely to impact the $^{10}$Be-$^{26}$Al inventory wherever they impinge on channel networks.

The dynamics of sediment transport, temporary storage and burial, are not easy to gauge through fluvial systems that are

many hundreds of kilometres long and, in places, tens of kilometres wide (Fig. 2). A few studies link the introduction of a burial signal in modern stream sediment to the reworking of alluvial sediment storages. Kober et al. (2009) suggest that in Rio Lluta, northern Chile, a downstream-increasing burial signal is potentially the result of reworked fluvial terraces (or slope and mass-wasting deposits) up to $10^5$ yr old. Similarly, Hidy et al. (2014) find that burial signals in streams on the coastal plain of

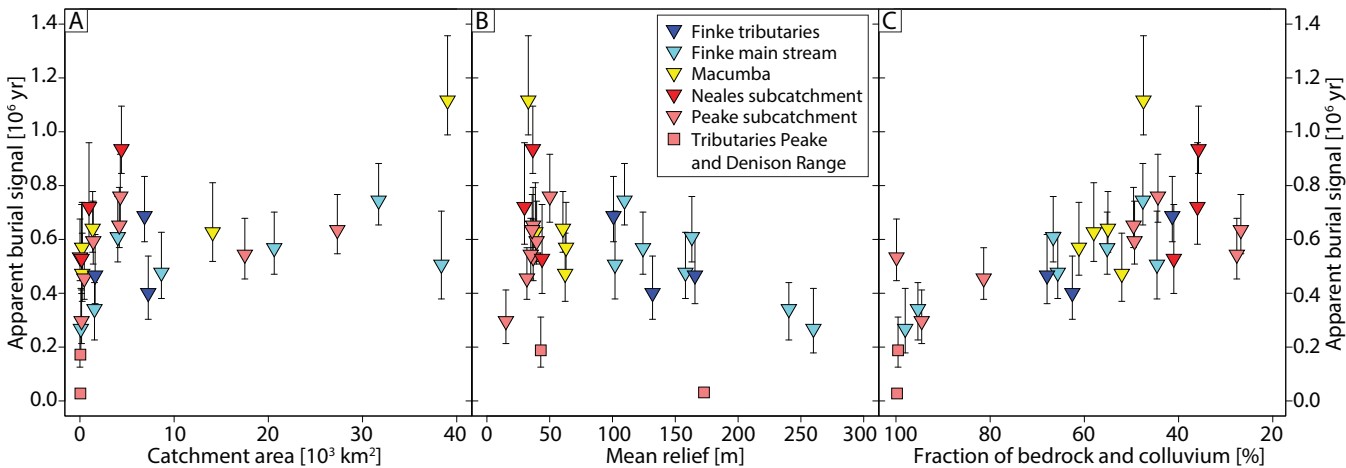

**Figure 8.** Apparent burial ages calculated for stream sediment – using CosmoCalc 3.0 (Vermeesch, 2007) – relative to A) drainage area, B) mean relief, C) fraction of exposed bedrock and colluvium cover. Finke samples are blue triangles (light blue – trunk stream; dark blue and white – tributaries), Macumba samples are yellow triangles, and Neales samples are red triangles and squares (dark – Neales subcatchment, light – Peake subcatchment, squares – Peake and Denison Range).

Texas stem from reworked pre- to mid-Pleistocene deposits. Bierman et al. (2005) identify that reworking long-buried (300–500 k.y.) floodplain material produces a burial signal in sediments of Rio Puerco, Colorado Plateau. Wittmann et al. (2011) detect Amazon floodplain burial signals in coarse (>500 μm) trunk-stream sediments sourced from reworked storages up to ~1.2 m.y. old. In central Australia, some useful guidance to minimum burial duration can be drawn from luminescence ages measured on shallow-buried fluvial sediments. Unlike [10]Be-[26]Al data, which can yield a cumulative burial signal, luminescence burial ages are reset by exposure to sunlight. Previously published TL ages from channel alluvium indicate minimum storage terms of >200 k.y. in the lower Neales (Croke et al., 1996) and >93 k.y. in the lower Finke (Nanson et al., 1995). Our three TL ages (Table A1) from the Macumba River floodplain depth-profile increase in age with depth, although the lowermost sample (160 cm) is saturated and therefore may be significantly older than the $120 \pm 9$ k.y. from 100 cm depth. Vertical accretion rates at these two floodplain sites span roughly ~8–54 mm k.y.$^{-1}$ and are compatible with the accretion rate of $64 \pm 33$ mm k.y.$^{-1}$ (mean $\pm 1$ σ) reported from Cooper Ck floodplain in the eastern Eyre Basin (Jansen et al., 2013). Of the 278 luminescence ages measured in Eyre Basin river sediments, mostly on Cooper Ck, one-third fall between 60–120 k.y. (the oldest being $740 \pm 55$ k.y.). Given the climatic and physiographic similarities between the eastern and western Eyre Basin, it seems reasonable to assume that minimum burial durations of $>10^5$ yr are representative of the Finke, Macumba, and Neales rivers. If a single storage interval may span $\sim 10^5$ yr, then it is feasible that the cumulative effect of many intervals of shallow-burial will cause the $^{26}$Al/$^{10}$Be ratio to deviate.

A similar argument applies to aeolian dune fields, which are major sediment-storages spanning ~3 million km$^2$ and up to 40% of the continent (Wasson et al., 1988; Hesse, 2010). All three catchments of the western Eyre Basin contain dunes in their lower reaches, but the Finke and Macumba have the strongest interaction in their lower reaches fringing the Simpson

Desert (Fig. 2). $^{26}$Al/$^{10}$Be burial ages suggest that dune accumulation probably began up to 1 m.y. ago (Fujioka et al., 2009) and, as with alluvial sediments, we infer minimum burial durations from luminescence dating. Based on a recent compilation listing 95 luminescence ages from the Simpson Desert (Hesse, 2016), minimum burial durations of >10$^5$ yr are widespread— the oldest dune sample yields a minimum age of 587 k.y. (Fujioka et al., 2009). In the hyper-arid Namib Desert, Bierman and

Caffee (2001) and Vermeesch et al. (2010) suggest that input of aeolian and/or reworked alluvium are responsible for decreased $^{26}$Al/$^{10}$Be ratios in modern sediments. Similar conclusions are drawn by Davis et al. (2012) for the Nile.

## 7 The $^{10}$Be-$^{26}$Al source-area signal in sediment-routing systems—a synthesis

### 7.1 Lithology drives heterogeneities in the source-area signal

Our comparison of $^{10}$Be measured in bedrock outcrops and hillslope soil, with $^{10}$Be in headwater streams reiterates the well-

known fact that source areas deliver highly diverse $^{10}$Be-$^{26}$Al inventories into stream networks, although the drivers of this diversity are less well understood. In rapidly-eroding mountain belts, the wide disparity in source-area erosion-rate ($10^2$–$10^3$ m m.y.$^{-1}$) is typically attributed to the effects of tectonism, such as seismicity and landsliding (Armitage et al., 2011). Yet, in central Australian streams, a comparable order of magnitude spread in source-area erosion rates ($10^{-1}$–$10^1$ m m.y.$^{-1}$) is chiefly due to lithology. Our data show that while $^{10}$Be-$^{26}$Al source-area signals are modified downstream (Fig. 7), disparities

in source-area erosion rates remain highly resilient. Despite hundreds of kilometres (∼200–600 km) of sediment mixing from source to sink, $^{10}$Be-$^{26}$Al inventories in western Eyre Basin streams (>1 km$^2$) retain a distinct signal of their source-area lithology (interquartile ranges): 0.2–0.4 m m.y.$^{-1}$ in the upper Peake (silcrete), 0.9–1.2 m m.y.$^{-1}$ in the Macumba (silcrete and granites), and 4.1–5.8 m m.y.$^{-1}$ in the Finke (quartzite-sandstone-conglomerate) (Fig. 4A; Table 3). This is consistent with the fundamental role that lithology plays in differentiating the tempo of erosion in all landscapes irrespective of their tectonic or

climatic setting (Scharf et al., 2013).

### 7.2 Are cosmogenic nuclide inventories reliable indicators of source-area erosion rate?

Estimates of catchment-scale erosion rate from cosmogenic nuclide abundances in sediment assume a high-fidelity relationship with the sediment source area (Bierman and Nichols, 2004; von Blanckenburg, 2005; Granger and Riebe, 2007; Dunai, 2010). However, as our data show, the down-system propagation of source-area signals tends to be scale-dependent: the widest spread

of $^{10}$Be occurs among hillslope bedrock outcrops (Fig. 5) from which the buffering effect of sediment transport downslope and downstream leads to progressively more stable catchment-averaged signals of erosion rate or particle burial (Wittmann and von Blanckenburg, 2016). This raises the question under what circumstances can we expect $^{10}$Be-$^{26}$Al inventories to yield an accurate picture of erosion in the source area. In the western Eyre Basin, the downstream shift in $^{26}$Al/$^{10}$Be ratio results in erosion-rate disparities (i.e. the difference between upstream and downstream samples) ranging from two-fold (Finke

and Macumba catchments) up to twelve-fold (Neales catchment) (Table 3). The validity of the assumption linking $^{10}$Be-$^{26}$Al

inventories to their source area reflects a systematic set of geomorphic conditions that requires consideration for reliable erosion rates to be obtained.

Source-area $^{10}$Be-$^{26}$Al inventories are largely unmodified in stream sediments traversing foreland basins fed by tectonically-active mountain belts, such as the Andes (Wittmann et al., 2009, 2011), the the Alps (Wittmann et al., 2016), and the Himalayas (Lupker et al., 2012; although no $^{26}$Al data are available here). Intermediate storage seems to have no appreciable effect on the low-$^{10}$Be source-area signal conveyed along these large, perennial, lowland rivers. Their sediment-routing systems are characterised by braiding channels leading on to anabranching and laterally-active meandering river styles—all indicative of high-discharge rivers optimised for sediment transfer. Frequent channel avulsion and fast lateral-migration rates bring channels into contact with older floodplain materials, but highly efficient reworking ensures a restricted age spread of sediments within the channel-belt and ongoing basin subsidence drives long-term sequestration into a rapidly thickening sediment pile (Allen, 2008; Armitage et al., 2011). In some cases, basin inversion may ultimately lead to recycling of older sediment storages back into the sediment-routing system, as shown in the upper Yellow River where reworked Neogene basin-fills alter the $^{26}$Al/$^{10}$Be source-area ratio downstream (Hu et al., 2011). From these examples, we can infer some key points favouring preservation of source-area signals: i) high sediment supply rates and therefore a channel-floodplain system configured for high sediment flux, ii) high mean runoff from headwaters, and iii) a thick sedimentary basin pile without older basin sediments exposed in the proximal floodplain/terraces.

The alternative limit case, in which the $^{10}$Be-$^{26}$Al source-area signal is modified downstream, follows distinctly different geomorphic conditions, summarised as: i) low sediment supply, and ii) juxtaposition of sediment storages with notably different exposure histories. Slow rates of source-area erosion (<20 m m.y.$^{-1}$) typical of low-relief postorogenic and shield-platform terrain (this study, Bierman et al., 2005; Hidy et al., 2014) produce down-system basin-fills that are thin and discontinuous. In the absence of subsidence creating accommodation space, there are juxtaposed sediment storages of widely differing age—and a high prospect of their admixture with the sediment-routing system (Kober et al., 2009; Davis et al., 2012; Hidy et al., 2014). Especially in dryland river systems, atmospheric inputs are typically part of a long-term history of fluvial-aeolian mass exchange (Bierman and Caffee, 2001; Bierman et al., 2005; Vermeesch et al., 2010; Davis et al., 2012). As described above, aeolian dune fields can host particles with notably longer exposure histories and burial timescales >1 m.y. (Fujioka et al., 2009; Vermeesch et al., 2010), and there is much observational evidence of fluvial-aeolian interactions in the western Eyre Basin.

## 8   Conclusions

We have tracked downstream variations in $^{10}$Be-$^{26}$Al inventories through three large sediment-routing systems (~100,000 km$^2$) in central Australia by comparing 56 cosmogenic $^{10}$Be and $^{26}$Al measurements in stream sediments with matching data (n = 55) from bedrock and soil mantles in the headwaters (Struck et al., 2018). Our summary conclusions are as follows:

1) Lithology is the primary determinant of erosion rate variations among bedrock outcrops in the order: silcrete, quartzite, sandstone, conglomerate (from slowest to fastest erosion rate). Our regional compilation of bedrock erosion-rate data yields interquartile ranges of 0.2–4.4 m m.y.$^{-1}$ on silcrete mesas in the Oodnadatta Tablelands; 1.6–4.8 m m.y.$^{-1}$ on quartzite-sandstone

ridges in the Western MacDonnell Ranges; 1.8–7.3 m m.y.$^{-1}$ on quartzite-sandstone in the Peake and Denison Range; and 6.7–6.8 m m.y.$^{-1}$ on conglomerate in the Western MacDonnell Ranges. Although $^{10}$Be-$^{26}$Al inventories are modified by sediment mixing over hundreds of kilometres downstream, they still retain a distinct signal of source-area lithology. Sediment-derived catchment-averaged erosion rates (interquartile ranges) are: 4.1–5.8 m m.y.$^{-1}$ for the Finke; 0.9–1.2 m m.y.$^{-1}$ for the Macumba; and 0.3–2.2 m m.y.$^{-1}$ for the Neales. The western headwaters of the Peake River (a subcatchment of the Neales River) yield 0.2–0.4 m m.y.$^{-1}$, which are among the slowest catchment-scale erosion rates ever measured (Table 3).

2) $^{10}$Be-$^{26}$Al inventories measured in stream-sediment samples from the Finke, Macumba, and Neales rivers all show overall downstream-increasing deviation from the steady-state erosion curve. These deviations correspond to minimum cumulative burial terms mostly between ∼400 and 800 k.y. (and up to ∼1.1 m.y.). The magnitude of the burial signal correlates with increasing sediment cover downstream (Figs. 7 and 8) and presumably results from assimilation of shallow-buried sediments from storages with long exposure histories, such as alluvial fans, desert pavements, floodplains and palaeo-alluvial plains, and aeolian dunes. In the lower reaches of the Peake and Neales rivers, the downstream-increasing burial signal is disrupted by inputs from faster-eroding landscapes in the Peake and Denison Range.

3) Downstream variations in $^{10}$Be-$^{26}$Al inventories weaken the fidelity of the relationship between source areas and catchment-averaged erosion-rate estimates from samples along large alluvial rivers. Based on our review of case studies that track $^{10}$Be-$^{26}$Al source-area signals downstream, we detect a set of behavioural trends under differing geomorphic settings. Preservation of source-area signals downstream is favoured by i) high sediment supply rates, ii) high mean runoff from headwaters, and iii) a thick sedimentary basin pile without older basin sediments exposed in the proximal floodplain. Conversely, source-area signals are more likely to be modified downstream in landscapes with: i) low sediment supply, and ii) juxtaposition of sediment storages with notably different exposure histories, such as aeolian dune fields. Such modifications can have significant impact on erosion rate estimates. In desert rivers of the western Eyre Basin, the downstream shift in $^{26}$Al/$^{10}$Be ratio results in erosion-rate disparities ranging from two-fold in the Finke and Macumba rivers, and up to twelve-fold in the Neales River (Table 3).

*Acknowledgements.* We thank Sarah Eccleshall for fieldwork assistance, Charles Mifsud for assistance with sample processing at ANSTO, and Jose Abrantes for conducting the XRD measurements at UOW. Financial support was provided by an Australian Research Council grant (DP130104023) to Gerald Nanson and JDJ, by a GeoQuEST Research Centre grant to JDJ and ATC, a Marie Skłodowska-Curie Fellowship to JDJ, and by the Centre for Accelerator Science at ANSTO through the National Collaborative Research Infrastructure Strategy. MS received an International Postgraduate Tuition Award provided by UOW and a matching scholarship funded by UOW and ANSTO. We acknowledge the Traditional Owners of this country.

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
