# Peer review of "Tracking the 10Be-26Al source-area signal in sediment-routing systems of arid central Australia"

_Earth Surface Dynamics, 2017_

## Referee Comment (RC1) · Anonymous Referee #1 · 1 Feb 2018

This manuscript "Tracking the 26Al/10Be source-area signal in sediment-routing systems of arid central Australia " by Struck et al., uses 26Al and 10Be inventories and ratios in bedrock, slope material and alluvial sediments to detect the sources of sediment into large rivers that flow into the western Eyre Basin. The authors conclude that in a setting of low relief topography, arid climate, and tectonic quiescence that characterize central Australia, the addition of stored sediments into the river system slowly changes the isotopic signature of the sediments such that they ultimately do not represent the erosional conditions at the source.

I think the paper is well written and interesting.

I do have a few comments:

1. Cosmogenic isotopes, such as 10Be, have been used as tracers for sediment routing at various temporal and spatial scales since the early 2000'ds. for example, in the Mojave Desert, in the Great Smoky Mountains, in the Negev desert. The authors should acknowledge this use and compare their conceptual results to previous ones.

2. The formation of desert pavements (Gibber) as described by Wells in 1995 and then demonstrated by Matmon et al., (2009) implies high Al and Be concentrations as well as ratios. The authors should include and consider that.

3. Please write 26Al-10Be when referring to the isotopic inventories and 26Al/10Be when referring to the isotopic ratios.

4. In regards to assumption ii about the discontinuity of sediment delivery. The discontinuity of sediment delivery is generally on an annual or decade scale. This temporal scale is obviously much shorter than the time scale measured by cosmogenic isotopes. Thus, in terms of cosmogenic isotopes the delivery of sediment is continuous. The authors should adders this point.

---

## Referee Comment (RC2) · Anonymous Referee #2 · 8 Feb 2018

My overall thoughts on this paper is that it is a well-organized, systematic, and mostly-clearly communicated study that has broad-arching implications for our understanding of how to approach and interpret cosmogenic data collected from quiescent continental interiors. I believe this study will make for an important contribution to geomorphology and will be very useful for those of us carrying out work in similar settings. I structure my review as suggested by the journal below:

1. This paper directly addresses relevant geomorphic questions, "How well do erosion signals at distal downstream sampling sites reflect source-area geomorphic processes in arid, low-relief, continental interiors?"

2. This paper comprises a large and important dataset of 29 new 10Be catchment-averaged erosion rates from an otherwise undersampled type of landscape: tectoni-

cally quiescent, low-relief, arid, continental interiors. The authors pair the 10Be concentrations used for deriving erosion rates with 26Al from the same samples to observe how the 10Be/26Al ratio is affected by geomorphic processes (sediment storage and admixture of stored sediment) as it is transferred from source to sink. The authors successfully synthesize multiple datasets of cosmogenic data and luminescence data from a variety of potential sediment sources to explore how environmental signals carried by eroded sediment are altered during sediment transport.

3. The authors reach a number of conclusions. Mainly, (1) that 10Be concentrations in distal downstream sediment reflect the 10Be inventories of their source area lithologies, (2) that burial signals from 10Be/26Al ratios increase downstream as rivers mix sediment from long-term storage sinks (dunes, fans, pavement, floodplain) with actively-transported fluvial sediment, and (3) that erosion rates derived from 10Be measurements in low-relief, arid, continental interiors with large amounts of sediment storage and episodic sediment transport should account for the cumulative burial signal prior to arriving at any conclusions about source-area landscape dynamics.

4. The scientific methods and assumptions are valid and clearly outlined. Moreover, the authors very clearly set up multiple working hypotheses and collect samples, which allow them to very clearly assess the viability of each hypothesis.

5. The results are sufficient to support the interpretations and conclusions made by the authors. Just as importantly, the manuscript is written well-enough, and the figures are drafted accordingly, that the reader is able to follow the authors' lines of reasoning easily.

6. The authors describe in sufficient detail the well-known and often-used technique of applying cosmogenic 10Be and 26Al measurements to address outstanding issues of geomorphic importance. Table 2 provides all necessary sample location information (coordinates, elevation), the sample mass, nuclide production scaling factors, 9Be and 17Al carrier masses, and the AMS 10Be/9Be and 26Al/27Al ratios. The manuscript

documents measurement and analytical uncertainties, the AMS standard materials used, the production rate used, the erosion rate derivation method used, the half-lives assumed, and the locations of sample measurement. Having never performed TL measurements myself, I am uncertain of how detailed the TL community prefers measurement details to be documented, but the authors cite appropriate methodological literature.

7. Yes. Very clearly up front, the authors cite numerous other studies that present findings showing how cosmogenic nuclide abundances can increase, decrease, or stay the same with distance from their source area. The authors also very clearly document important studies that have enhanced our understanding of what cosmogenic nuclides measured from stream sediment in rivers that drain active mountain belts tell us about source area dynamics, and the authors clearly depict in Figure 1 how their work will improve our understanding of what environmental signals are carried by CRNs in rivers draining low-relief, inactive, continental interiors.

8. Yes. The title aptly describes the content of the paper.

9. Yes. The Abstract is concise and complete. A few comments here, though. Line 4 of Abstract mentions "the factors responsible" but the authors do not say responsible for what? The reader is left unsure. Line 10 of the Abstract mentions "downstream-increasing minimum cumulative burial terms," which is a mouthful and should be simplified considering the authors do not focus much in the paper itself on the notion that burial times are "minimum-cumulative."

10. Generally, yes. This is a fantastic example of a clear, direct, and easy to follow research study. I was able to mark exactly where the authors state (1) the knowledge gap, (2) the working models, (3) the hypotheses being tested, (4) key results, (5) significance of the results, and (6) why the field area was able to address the questions being asked. I did find the thermoluminescence methods seemingly out of place, since nowhere prior to Section 3.2 was it mentioned that TL was part of this study, and while

the TL results are important for interpreting observations (Page 9, Line 34 – Page 10, Lines 1-3), they seemed to come out of nowhere and disappear again until the Discussion. I suggest that the authors more clearly state why TL dating was needed up front and incorporate the methods more seamlessly into the manuscript.

11. The language was fluent and precise. However, I found myself confused between the use of the word "downstream" throughout the manuscript and the presentation of data in Figures 5 and 7 (middle panels), which was plotted as "distance upstream." It took me a while to realize that by reading each of the panel-sets in Figures 5 and 7 from Left to Right, I was following either the 10Be inventory decrease or the burial signal increase from source to sink. That could be made clearer in the manuscript text to help the reader follow the clever way the data were presented. The same confusion came with the plots of the "fraction of bedrock and colluvium" because 100% was on the left-hand side of the x-axis. Again, this is because there is a greater % of bedrock in the source area, but it took me a few attempts at reading the figures before I grasped why the authors presented their data in this way. In sum, I think the authors should be more specific in the manuscript or figure caption to help the reader digest their well-thought-out plots.

12. Mathematical formulae: There were no formulae presented in the manuscript. Symbols: In Figure 1, "Q" is used but never defined. Please define. Abbreviations: I noticed no confusing abbreviations. Units: I notice that the nomenclature of some units is inconsistent throughout. For instance "atoms g-1" compared to "m/M.y."

13. Clarifications: 13a. P.1, L.4-5: "…identify the factors responsibe." Responsible for what?

13b. P.1, L.20-21: Sentence starting with "The timescale…" I think the wording here is fundamentally backwards. Sediment transfers occur at their own pace, regardless of how we measure it. But how we measure it determines the timescales over which we can make inferences about sediment transfers. I agree with the following sentences,

but please consider rewording here.

13c. P.4, L.5-8: Have the authors considered how such long-wavelength deformation might affect CRN inventories?

13d. P.4, L.21-26: This is repeated from further up in the section. Please consolidate.

13e. P.5, L.2: "...respectively." Which classes were assigned to which domains? It does appear to be clear. Please revise for clarification.

13f. P.6, L.3: Section 4.1 only addresses 10Be abundances, and does not mention 26Al. I suggest the authors either detail how the 26Al abundances change with sample position downstream as they do with 10Be (difficult for a reader to do this by looking at the table only), or remove "26Al" from section header.

13g. P.6, L.25-26: Considering that one of the main points of this study is to demonstrate how incorporation of buried sediment in downstream catchments affects interpretation of upstream erosion rates, and considering that the authors suggest a 2-12x change in erosion rates after burial is accounted for, the range of erosion rates shown here appears to show a less-severe change to modeled erosion rates. In fact, I look at Table 3, and I cannot find a sample from the Neales catchment that shows a 12-fold disparity in erosion rate before/after burial is accounted for (as is said on P.11, L11). This is a point I feel that the authors must make more clear or obvious to the reader prior to publication.

13h. P.6 L.28: Section 5.1. I would really like to see a figure that clearly shows the similarities between fluvial sediment inventories and bedrock inventories. Given that this is the authors' primary conclusion, I find it odd that this is not more clearly shown.

13i. P.7, L.20-26: I find the wording here to be inconsistent with the values presented. The authors state that (1) the ridges and hillslopes of the MacDonnell Ranges have similar and low 10Be abundances, (2) bedrock feeds headwater streams directly, but then (3) headwater streams exhibit a wide range of 10Be inventories, which reflects

bedrock composition. How can the ridges have similar and low 10Be abundances while at the same time feed a very wide range of 10Be directly to the streams? This is where the conglomerate and the quartzite-sandstone come from, which have distinct 10Be abundances, so I agree with the authors that the wide range in 10Be inventories reflects this, but then the ridges of the MacDonnell Ranges cannot have similar and low 10Be inventories. This seems easily resolved by rewording and clarification.

13j. P.7, L.28-29: The authors suggest that a rise in 10Be amount coincides with a shrinking fraction of bedrock and growing fraction of sediment cover, implying slight burial/storage? But in this case, couldn't an increase in 10Be also signify a decreasing legacy of the rapidly eroding conglomerate to the increasing signal from the more slowly eroding quartzite-sandstone from bedrock and hillslope materials?

13k. P.9, L.5: Not sure why Portenga & Bierman (2011) is cited here. I'd remove or clarify.

13l. P.11,L.11: The authors here mention a 12-fold disparity in erosion rates if burial is or is not accounted for in the Neales catchment. I still do not understand where this 12-fold statistic is coming from. I see on Figure 7 that there is a 12-fold change in the apparent burial signal from the 10Be/26Al ratios for both the Macumba and Peakes/Neales catchments, but the erosion values in Table 3 are not 12-times greater or less after burial is accounted for than when burial is not accounted for. Perhaps I am not understanding the authors' interpretations here, but I think this requires clarification prior to publication.

13m. Figure 2 shows the locality of Oodnadatta and they refer to the Oodnadatta Tablelands in the manuscript, but I am not sure if the Oodnadatta Tablelands are the low elevation area around the locality of Oodnadatta, or if the Tablelands are formed by the silcrete mesas? Please clarify, or show with words on the map. Also in Figure 2, the Musgrave Range is labeled, but never mentioned in the text, but its inclusion made me wonder how sediment sourced from the Musgrave Range might impact cosmogenic

data from the Macumba River catchment?

13n. Figure 3: Great images. Panel A should have a N-arrow on it since Figure 2 suggests the range strikes E-W, but it is shown up-photo-down-photo here. Panel C inset should have a scale bar; at first glance, I thought this was the sand collected, not the pebbles of the desert pavement.

13o. Figure 5: Please label each of the figure subsets with the catchment name. I found myself flipping back and forth in order to remember which catchment was which. Or maybe use the same map insets you use for Figure 6? Lastly, the authors say that Figures 5A, D, and G show apparent burial ages, but only 10Be concentration is shown, not ages. Wording of the text should change to be consistent.

13p. Figure 7: Please label each subset of figures with the catchment name.

14. The number and quality of references is appropriate.

15. Amount and quality of the supplementary material is helpful and appropriate.

---

## Short Comment (SC1) · 8 Feb 2018

This paper deals with cosmogenic nuclids insight into either denudation and sediment routing in a large arid area. It brings very interesting ideas well supported by the data. I feel it must be published for that.

I have one comment regarding denudation. The paper indicates higher erosion rates in the lower parts of the catchment. Flat topped hills made of silcrete erode slower. Thus maybe the landscape 'starting state' was a flat peneplain currently carving at a slow rate. If this idea is correct, to what period the starting point could correspond? is there any scenario for 'reactivation' of this landscape (I remember the area is the locus for Neogene marine terraces/deposits)?

[Figure]

This is a nice work.

Vincent Regard, Toulouse, France
* * *

---

## Author Comment (AC1)

**Response to Reviews**

**Struck et al. - *Tracking the $^{10}$Be-$^{26}$Al source-area signal in sediment-routing systems of arid central Australia***

Here we respond to referee comments and provide explanations of our revisions. We thank the referees for their constructive and supportive suggestions. We have by and large followed them all. Referee comments are copied in grey and marked RC plus a sequential number. Our responses are marked R and keyed to page P and line L.

**Referee #1 (Comments to the author):**

| | |
|---|---|
| RC1 | Cosmogenic isotopes, such as $^{10}$Be, have been used as tracers for sediment routing at various temporal and spatial scales since the early 2000'ds. for example, in the Mojave Desert, in the Great Smoky Mountains, in the Negev desert. The authors should acknowledge this use and compare their conceptual results to previous ones. |
| R1 | Agreed, cosmogenic isotopes have been used very widely as tracers for sediment routing, and so any citation list must be selective. In addition to the pioneering studies (i.e., McKean et al., 1993; Bierman and Steig, 1995; Brown et al., 1995; Granger et al., 1996), we had already listed what we regard as a couple of highly innovative studies (i.e., Heimsath et al., 2005 and Anderson, 2015). To these we have now added some notable field-based studies: Nichols et al. (2002); Matmon et al. (2003); and Jungers et al. (2009). (P.2 L.21)

The source-to-sink conceptual framework used in this MS links with our accompanying paper (Struck et al., 2018, GSAB) that deals with the hillslope system. Based on these two substantial field studies (including 117 $^{10}$Be and $^{26}$Al measurements), we have placed our conceptual results firmly in the context of previous work by proposing two limit cases in which source-area nuclide inventories are modified or not by the sediment-routing system. We have then considered and discussed numerous controls pertinent to our field sites that might govern such variations. |
| RC2 | The formation of desert pavements (Gibber) as described by Wells in 1995 and then demonstrated by Matmon et al., (2009) implies high Al and Be concentrations as well as ratios. The authors should include and consider that. |
| R2 | We have now cited the two key studies noted above in the following:
'Long residence times and slow hillslope evolution arise from the lack of fluvial incision associated with widespread base-level stability and the long-lasting development of stony soil mantles, also known as desert pavement (Mabbutt, 1977; Wells et al., 1995, Fujioka et al., 2005, Matmon et al., 2009).' [P.4, L.32 – P.7, L.1]

In our previous paper (i.e., Struck et al., 2018), we examine the sediment production and age dynamics of desert pavements developed on hillslopes by measuring $^{10}$Be and $^{26}$Al at three separate sites in the western Eyre Basin. In our Supplementary Table A3, we reproduce the nuclide data from Struck et al. (2018). This shows gibber samples with very high $^{10}$Be abundances (up to 5.4 M atoms g$^{-1}$) and $^{26}$Al abundances (up to 18.7 M atoms g$^{-1}$), as suggested by the reviewer and consistent with the key studies by Wells and Matmon among others. However, we find rather low $^{26}$Al/$^{10}$Be ratios at two of the three hillslope sites, with samples that plot well below the steady-state erosion island. We propose some new explanations for these findings that stem from extremely slow denudation and we refer the reviewer to our other paper for details. Here, our aim is to draw together our knowledge of source-area nuclide inventories established in that previous paper with inventories we measured in the fluvial systems downstream. |
| RC3 | Please write $^{26}$Al-$^{10}$Be when referring to the isotopic inventories and $^{26}$Al/$^{10}$Be when referring to the isotopic ratios. |

| | |
|---|---|
| R3 | To clarify the issue further, we have modified the text to $^{10}$Be-$^{26}$Al when referring to the nuclide inventories and the nuclide signal in general, and we now use $^{26}$Al/$^{10}$Be only when referring specifically to the nuclide ratio. |
| RC4 | In regard to assumption ii about the discontinuity of sediment delivery. The discontinuity of sediment delivery is generally on an annual or decade scale. This temporal scale is obviously much shorter than the time scale measured by cosmogenic isotopes. Thus, in terms of cosmogenic isotopes the delivery of sediment is continuous. The authors should address this point. |
| R4 | The sediment flux is indeed continuous on the timescale integrated by $^{10}$Be and $^{26}$Al. We have now changed text in three places to correct this misconception regarding the $^{10}$Be-$^{26}$Al source-area signal:
 i) We have removed reference to the discontinuity of sediment flux in the Abstract (P.1, L.15) and the Conclusions (P.25, L.19).
 ii) We have also removed reference to the discontinuity due to ephemeral stream-flows at the end of the Discussion (P.24 L.18), but we have kept the section regarding fluvial-aeolian interactions, as follows:
 'Especially in dryland river systems, atmospheric inputs are typically part of a long-term history of fluvial-aeolian mass exchange (Bierman and Caffee, 2001; Bierman et al., 2005; Vermeesch et al., 2010; Davis et al., 2012). As described above, aeolian dune fields can host particles with notably longer exposure histories and burial timescales >1 m.y. (Fujioka et al., 2009; Vermeesch et al., 2010), and there is much observational evidence of fluvial-aeolian interactions in the western Eyre Basin.' (P.24, L.23-26). |

**Referee #2 (Comments to the author):**

| | |
|---|---|
| RC5 | Line 4 of Abstract mentions "the factors responsible" but the authors do not say responsible for what? The reader is left unsure. |
| R5 | We have now modified this sentence to:
 '… with the aim of tracking downstream variations in $^{26}$Al-$^{10}$Be inventories and to identify the factors responsible for these variations.' (P.1, L.4-5) |
| RC6 | Line 10 of the Abstract mentions "downstream-increasing minimum cumulative burial terms," which is a mouthful and should be simplified considering the authors do not focus much in the paper itself on the notion that burial times are "minimum-cumulative." |
| R6 | i) We have now modified the relevant sentence to:
 '$^{10}$Be-$^{26}$Al inventories in stream-sediments indicate that cumulative-burial terms increase downstream to mostly ~400–800 k.y. and up to ~1.1 m.y.' (P.1, L.9-11)
 ii) As noted in RC6, the burial terms are minimum-cumulative, but we agree that it is sufficient to clarify this point in the main text, which reads:
 'These deviations correspond to minimum cumulative burial terms mostly between ~400 and 800 k.y. (and up to ~1.1 m.y.).' (P.25, L.8-9) |
| RC7 | I did find the thermoluminescence methods seemingly out of place, since nowhere prior to Section 3.2 was it mentioned that TL was part of this study, and while the TL results are important for interpreting observations (Page 9, Line 34 – Page 10, Lines 1-3), they seemed to come out of nowhere and disappear again until the Discussion. I suggest that the authors more clearly state why TL dating was needed up front and incorporate the methods more seamlessly into the manuscript. |
| R7 | i) We have now added a phrase to the end of the Introduction to introduce the albeit minor role played by luminescence dating in our study:
 '... (Struck et al., 2018), and we supplement those with four thermoluminescence ages on floodplain sediments' (P.3, L.13 – P.4, L.1)
 ii) We have now added a short rationale for the TL dating to the Methods (Section 3.2):
 'With the aim of gauging the burial age of floodplain sediments flanking some of our study |

| | |
|---|---|
| | channels, we collected four samples for thermoluminescence (TL) dating ...' (P.8, L.9-10)
iii) As we note in the Discussion (P.22, L.6-16), storage terms of $10^4$-$10^5$ y have been demonstrated elsewhere in the Eyre Basin by previous workers (Nanson et al., 1995; Croke et al., 1996) based on hundreds of luminescence measurements. We compare our new TL results with the published data in the same section of the Discussion. The inclusion of TL in our study therefore confirms that alluvial sediment storage in the western Eyre Basin fits within the larger regional context. |
| RC8 | However, I found myself confused between the use of the word "downstream" throughout the manuscript and the presentation of data in Figures 5 and 7 (middle panels), which was plotted as "distance upstream." It took me a while to realize that by reading each of the panel-sets in Figures 5 and 7 from Left to Right, I was following either the $^{10}$Be inventory decrease or the burial signal increase from source to sink. That could be made clearer in the manuscript text to help the reader follow the clever way the data were presented. The same confusion came with the plots of the "fraction of bedrock and colluvium" because 100% was on the left-hand side of the x-axis. Again, this is because there is a greater % of bedrock in the source area, but it took me a few attempts at reading the figures before I grasped why the authors presented their data in this way. In sum, I think the authors should be more specific in the manuscript or figure caption to help the reader digest their well-thought-out plots. |
| R8 | We aimed to show our data following the overall source-to-sink concept of the MS. We have now reworked Figs. 5 and 7 captions overall to make them more readable.
i) To both captions we have added:
'—note that we have reversed the x-axes in all panels to illustrate our data from source-to-sink, left to right.'
ii) In both captions we have changed the phrase from 'upstream distance to lowermost sample' to 'distance along-stream from most downstream sample'. And we have changed the x-axis label in the Figures accordingly. |
| RC9 | In Figure 1, "Q" is used but never defined. Please define. |
| R9 | We have now defined 'Qs' as sediment flux in the Fig. 1 caption. |
| RC10 | I notice that the nomenclature of some units is inconsistent throughout. For instance, "atoms g-1" compared to "m/M.y." |
| R10 | We have now replaced m/m.y., mm/yr, and mm/k.y. with m m.y.$^{-1}$, mm yr$^{-1}$, and mm k.y.$^{-1}$ and maintained the exponential form of units throughout the MS. |
| RC11 | P.1, L.4-5: "…identify the factors responsible." Responsible for what? |
| R11 | Please see R5 above. |
| RC12 | P.1, L.20-21: Sentence starting with "The timescale…" I think the wording here is fundamentally backwards. Sediment transfers occur at their own pace, regardless of how we measure it. But how we measure it determines the timescales over which we can make inferences about sediment transfers. I agree with the following sentences, but please consider rewording here. |
| R12 | The sentence in question is not necessary; we have now removed it. (P.1, L.18) |
| RC13 | P.4, L.5-8: Have the authors considered how such long-wavelength deformation might affect CRN inventories? |
| R13 | Some of the co-authors have studied the effects of deformation on nuclide inventories in the Eyre Basin. Given the low rates of surface uplift and erosion, the effects are not well defined although subsidence linked to active synclinal structures in the eastern Eyre Basin has formed sediment traps that increase sediment storage on $10^3$–$10^6$ timescales (Jansen et al. 2013). Nevertheless, the nuclide inventories are still dominated by the lithological signal. |
| RC14 | P.4, L.21-26: This is repeated from further up in the section. Please consolidate. |
| R14 | We agree that there is some repetition: the overall aims are set out at the close of the Introduction, as is conventional. But we then briefly reiterate some aspects more specifically in a way relevant to the Methods that immediately flow. This seems well placed and useful in |

| | |
|---|---|
| | our view for keeping the reader informed as to what is coming next. |
| RC15 | P.5, L.2: "…respectively." Which classes were assigned to which domains? It does appear to be clear. Please revise for clarification. |
| R15 | We have now clarified the text as follows:
'Bedrock and depositional landforms were sorted into seven different classes: exposed bedrock (no silcrete), exposed silcrete, colluvium cover, gibber cover (desert pavement), aeolian cover, sand plains, and alluvium. Of this group, the first three classes were assigned to the bedrock-hillslope domain and the latter four were assigned to the sediment cover domain.' (P.7, L.19-21). |
| RC16 | P.6, L.3: Section 4.1 only addresses [10]Be abundances, and does not mention [26]Al. I suggest the authors either detail how the [26]Al abundances change with sample position downstream as they do with [10]Be (difficult for a reader to do this by looking at the table only), or remove "[26]Al" from section header. |
| R16 | We have now removed [26]Al from the section header (P.13, L.9). |
| RC17 | P.6, L.25-26: Considering that one of the main points of this study is to demonstrate how incorporation of buried sediment in downstream catchments affects interpretation of upstream erosion rates, and considering that the authors suggest a 2-12x change in erosion rates after burial is accounted for, the range of erosion rates shown here appears to show a less-severe change to modeled erosion rates. In fact, I look at Table 3, and I cannot find a sample from the Neales catchment that shows a 12-fold disparity in erosion rate before/after burial is accounted for (as is said on P.11, L11). This is a point I feel that the authors must make more clear or obvious to the reader prior to publication. |
| R17 | The 2 to 12-fold erosion-rate disparity is based on the difference between the upstream and downstream samples after accounting for burial. For example, a 12-fold erosion-rate change occurs on the Neales between PEA4 and PEA8. To clarify this misunderstanding, we have noted exactly what we mean by the disparity:
'… the downstream shift in [26]Al/[10]Be ratio results in erosion-rate disparities (i.e., the difference between upstream and downstream samples) ranging from two-fold (Finke and Macumba catchments) up to twelve-fold (Neales catchment) (Table 3).' (P.23, L.28-30). |
| RC18 | P.6 L.28: Section 5.1. I would really like to see a figure that clearly shows the similarities between fluvial sediment inventories and bedrock inventories. Given that this is the authors' primary conclusion, I find it odd that this is not more clearly shown. |
| R18 | A comparison between the sediment and bedrock nuclide inventories is shown figuratively in three different ways (in addition to the cosmogenic nuclide data reported in Table 2), as follows:
i) In Fig. 5, we show the [10]Be abundances measured in bedrock samples and hillslope soil samples (panels A, D, G) compared directly to the fluvial sediment samples (panels B, E, H). These data, which include previously published results (from Struck et al., 2018, Heimsath et al., 2010, and Fujioka et al., 2005) show a considerable overlap together with a strong lithological signal, as we discuss (P.18, L.23 – P.21, L.3).
ii) In Fig. 6, we use two-nuclide plots to show and compare the [10]Be-[26]Al inventories measured in bedrock samples, hillslope soil samples, and fluvial sediment samples.
iii) In Fig. 7, we show and compare the apparent burial ages calculated for bedrock samples and hillslope soil samples (panels A, D, G), and fluvial sediment samples (panels B, E, H).

We conducted a more comprehensive comparison of the [10]Be-[26]Al inventories measured in bedrock and hillslope soil in our previous paper in GSAB, Struck et al. (2018), which we cite in section 5.1. We now refer here to Figure 13 of this GSAB paper which demonstrates the lithological influence on erosion rates in the region.(P.18, L.6) |
| RC19 | P.7, L.20-26: I find the wording here to be inconsistent with the values presented. The authors state that (1) the ridges and hillslopes of the MacDonnell Ranges have similar and |

| | |
|---|---|
| | low [10]Be abundances, (2) bedrock feeds headwater streams directly, but then (3) headwater streams exhibit a wide range of 10Be inventories, which reflects bedrock composition. How can the ridges have similar and low [10]Be abundances while at the same time feed a very wide range of [10]Be directly to the streams? This is where the conglomerate and the quartzite-sandstone come from, which have distinct [10]Be abundances, so I agree with the authors that the wide range in [10]Be inventories reflects this, but then the ridges of the MacDonnell Ranges cannot have similar and low [10]Be inventories. This seems easily resolved by rewording and clarification. |
| R19 | We have now clarified this issue by rephrasing the section 5.2 opening to: 'The prominent strike ridges and hillslope soil mantles of the MacDonnell Ranges (Fig. 3A) contain a wide range of abundances of [10]Be ~0.2–6.5 x $10^6$ atoms g[-1] (Fig. 5A), which appears to be driven by bedrock lithology (cf. Fig. 13 in Struck et al, 2018). In some cases, small alluvial fans form intermediate storages of sediment prior to it entering the stream network, but more commonly bedrock ridges feed sediment directly to low-order headwater streams (Fig. 5B). High [10]Be (1–5 x $10^6$ atoms g[-1]) occurs in streams draining resistant quartzite ridges, whereas streams from sandstone-siltstone ridges and low conglomerate hills yield ~0.3–0.6 x $10^6$ atoms g[-1].' (P.18, L.24-29).

 We now show the full ranges of bedrock [10]Be abundances (including previously published data) in Fig. 5 A, D, and G, and we present the previously published data in Table A3. |
| RC20 | P.7, L.28-29: The authors suggest that a rise in [10]Be amount coincides with a shrinking fraction of bedrock and growing fraction of sediment cover, implying slight burial/storage? But in this case, couldn't an increase in [10]Be also signify a decreasing legacy of the rapidly eroding conglomerate to the increasing signal from the more slowly eroding quartzite-sandstone from bedrock and hillslope materials? |
| R20 | The suggested scenario of a decreasing legacy of fast-eroding conglomerate does not apply, because the samples with low [10]Be (B123s, F1, and PIO) do not contain conglomerate in their catchments, and nor do the other six samples H8-H37 shown in Fig. 5A and B. The conglomerate terrain drainage enters the Finke River between samples F1 and F2 where it appears to exert minor effect (Fig. 5B). |
| RC21 | P.9, L.5: Not sure why Portenga & Bierman (2011) is cited here. I'd remove or clarify. |
| R21 | We have now removed the citation (P.21, L.8). |
| RC22 | P.11,L.11: The authors here mention a 12-fold disparity in erosion rates if burial is or is not accounted for in the Neales catchment. I still do not understand where this 12-fold statistic is coming from. I see on Figure 7 that there is a 12-fold change in the apparent burial signal from the [10]Be/[26]Al ratios for both the Macumba and Peakes/Neales catchments, but the erosion values in Table 3 are not 12-times greater or less after burial is accounted for than when burial is not accounted for. Perhaps I am not understanding the authors' interpretations here, but I think this requires clarification prior to publication. |
| R22 | Please see R17 above. |
| RC23 | Figure 2 shows the locality of Oodnadatta and they refer to the Oodnadatta Tablelands in the manuscript, but I am not sure if the Oodnadatta Tablelands are the low elevation area around the locality of Oodnadatta, or if the Tablelands are formed by the silcrete mesas? Please clarify, or show with words on the map. |
| R23 | Indeed, the Oodnadatta Tablelands is the region containing the silcrete plateaus and mesas in the western part of the Neales. We have now clarified this by adding a label to Fig. 2. |
| RC24 | Also in Figure 2, the Musgrave Range is labeled, but never mentioned in the text, but its inclusion made me wonder how sediment sourced from the Musgrave Range might impact cosmogenic data from the Macumba River catchment? |
| R24 | Our Alberga River samples (ALB1–3) must contain input from the Musgrave Ranges; however, we cannot gauge the effects of these uplands because we were unable to sample this very |

| | |
|---|---|
| | remote area. |
| RC25 | Figure 3: Great images. Panel A should have a N-arrow on it since Figure 2 suggests the range strikes E-W, but it is shown up-photo-down-photo here. Panel C inset should have a scale bar; at first glance, I thought this was the sand collected, not the pebbles of the desert pavement. |
| R25 | We have now added a N-arrow to panel A and a scale-bar to panel C. |
| RC26 | Figure 5: Please label each of the figure subsets with the catchment name. I found myself flipping back and forth in order to remember which catchment was which. Or maybe use the same map insets you use for Figure 6? Lastly, the authors say that Figures 5A, D, and G show apparent burial ages, but only $^{10}$Be concentration is shown, not ages. Wording of the text should change to be consistent. |
| R26 | We have now added map insets and labels to Fig. 5 (as in Fig. 6), and we have corrected the wording in the Fig. 5 caption. |
| RC27 | Figure 7: Please label each subset of figures with the catchment name. |
| R27 | We have now added map insets and labels to Fig. 7 (as in Fig. 6). |

Short Comment #1 (Comments to the author):

| | |
|---|---|
| SC28 | The paper indicates higher erosion rates in the lower parts of the catchment. Flat topped hills made of silcrete erode slower. Thus maybe the landscape 'starting state' was a flat peneplain currently carving at a slow rate. If this idea is correct, to what period the starting point could correspond? Is there any scenario for 'reactivation' of this landscape (I remember the area is the locus for Neogene marine terraces/deposits)? |
| R28 | Based on $^{10}$Be-$^{21}$Ne measurements on gibbers derived from silcrete mesas in the Oodnadatta Tablelands, inheritance-corrected exposure ages suggest dissection started at ~2-4 Ma (Fujioka et al., 2005, Geology 33, 993-996). However, there is no evidence that the silcrete duricrust ever formed a continuous erosion surface (or 'peneplain'), as proposed by Woolnough (1927). That idea is now discredited. Some of the silcretes are pedogenic, suggesting near-surface genesis, and others are groundwater silcretes and therefore not directly associated with a former ground surface. For a Davisian perspective on the landscape evolution of the western Eyre Basin, see Simon-Coinçon et al. 1996, J. Geol. Soc. Lond. 153, 467-480. Silica-enrichment probably occurred at topographic low points and formation was very often time-transgressive. For an excellent review, see Taylor & Eggleton 2017, Aust. J. Earth Sci. 64, 987-1016. |